# Plastic waste discharge to the global ocean constrained by seawater observations

Yanxu Zhang [1,2,11] ✉, Peipei Wu[1,11], Ruochong Xu[1], Xuantong Wang[1], Lili Lei [1] ✉, Amina T. Schartup [3], Yiming Peng[1], Qiaotong Pang[1], Xinle Wang[1], Lei Mai [4], Ruwei Wang[4], Huan Liu [5], Xiaotong Wang [5], Arjen Luijendijk[6,7], Eric Chassignet [8], Xiaobiao Xu [8], Huizhong Shen[9], Shuxiu Zheng[10] & Eddy Y. Zeng [4] ✉

Marine plastic pollution poses a potential threat to the ecosystem, but the sources and their magnitudes remain largely unclear. Existing bottom-up emission inventories vary among studies for two to three orders of magnitudes (OMs). Here, we adopt a top-down approach that uses observed dataset of sea surface plastic concentrations and an ensemble of ocean transport models to reduce the uncertainty of global plastic discharge. The optimal estimation of plastic emissions in this study varies about 1.5 OMs: 0.70 (0.13−3.8 as a 95% confidence interval) million metric tons yr$^{-1}$ at the present day. We find that the variability of surface plastic abundance caused by different emission inventories is higher than that caused by model parameters. We suggest that more accurate emission inventories, more data for the abundance in the seawater and other compartments, and more accurate model parameters are required to further reduce the uncertainty of our estimate.

Marine plastic debris has become a growing concern in recent years due to its potential threat to wildlife and humans[1,2]. Ocean plastic waste mainly derives from terrestrial sources via either riverine discharge or the erosion of waste from coastal zones while a smaller contribution is from direct dumping by shipping and fishing activities (-25% of the total terrestrial discharge)[3–5]. A majority of these sources ultimately result from the mismanaged plastic waste (MPW) associated with soaring plastic production in the past 70 years[6]. However, the global flux of plastic waste emitted to the ocean remains unclear, which severely hinders our ability to make effective mitigation strategies.

Estimating the emissions of plastic debris to the global ocean is a challenging task due to the large spatiotemporal heterogeneity of MPW generations and hydrological conditions of the river watersheds and coastal areas[4,5]. One limitation is the data availability as we have only -10² rivers with data so far, compared to the large number (>10⁶) of all the rivers worldwide[7,8]. Regression models are often developed between the measured discharge and proxy data, such as population, plastic use, MPW generation, income level, land use, runoff, and precipitation[5,7–9]. These models are then applied to other rivers without measurement data to achieve a global estimate. Using different

[1]School of Atmospheric Sciences, Nanjing University, 210023 Nanjing, China. [2]Frontiers Science Center for Critical Earth Material Cycling, Nanjing University, 210023 Nanjing, China. [3]Scripps Institution of Oceanography, University of California, San Diego, La Jolla, CA, USA. [4]Center for Environmental Microplastics Studies, Guangdong Key Laboratory of Environmental Pollution and Health, School of Environment, Jinan University, 511443 Guangzhou, China. [5]State Key Joint Laboratory of ESPC, State Environmental Protection Key Laboratory of Sources and Control of Air Pollution Complex, School of Environment, Tsinghua University, Beijing, China. [6]Faculty of Civil Engineering and Geosciences, Delft University of Technology, Delft, Netherlands. [7]Hydraulic Engineering, Deltares, Delft, Netherlands. [8]Center for Ocean–Atmospheric Prediction Studies (COAPS), Florida State University, Tallahassee, FL, USA. [9]School of Environmental Science and Technology, Southern University of Science and Technology, Shenzhen, Guangdong, China. [10]College of Urban and Environmental Sciences, Peking University, Beijing, China. [11]These authors contributed equally: Yanxu Zhang, Peipei Wu. ✉e-mail: zhangyx@nju.edu.cn; lililei@nju.edu.cn; eddyzeng@jnu.edu.cn

subsets of rivers and choosing different proxy data, the estimated global flux can range by up to three orders of magnitudes (OMs) among these studies (Fig. 1). Also, rivers act as reservoirs for plastics, affecting the discharge to oceans[10]. There is a similar or even worse paucity of data for the emissions from the erosion of waste from coastal zones[4].

In contrast to directly measured emissions, surface ocean plastic abundances have been extensively measured in the recent decade. There are more than 3000 samples of surface net tow data, mostly located in the centers of gyres in the northern hemisphere that are found to be plastic garbage patches[11]. Previous studies summarized these data and estimated a load of global plastic in the surface ocean, which ranges 14.4–236 thousand metric tons (Mts)[11–13]. However, these estimates are not directly comparable with plastic emissions, given the complex transport and transformation processes of plastic particles in the ocean (e.g., windage, photo-degradation, beaching, fragmentation, sinking and buoying, and biofouling/defouling)[14]. A large fraction of the emissions are removed from the surface ocean by these processes and is known as "missing plastics"[11,15]. Numerical models with a variety of complexity have been developed in the past decade[16–22]. These models track the spatial pattern relatively well but contain substantial uncertainties of 2–3 OMs in estimating the surface ocean plastic abundance[10].

In this study, we use a top-down approach to estimate the plastic waste discharge to the global ocean based on surface ocean plastic abundance data. A similar method has been adopted by Brahney et al. to constrain the atmospheric sources of plastics[23]. We develop a three-dimensional Euler-based global ocean plastic model, which explicitly includes a comprehensive representation of the important processes for plastic particles in the ocean (e.g., sinking and rising, drifting, fragmentation/abrasion, beaching, and biofouling/defouling). The model is driven by ocean physics data from an ocean general circulation model and simulates the historical emissions, transport, transformation, and accumulation of different types and sizes of plastic particles from 1950 to 2018 (see the "Methods" section). The model results are constrained by measured surface ocean plastic abundances sampled between 2000 and 2015. We limit the observations to the data obtained by visual identification due to its large sample size and spatial coverage (Supplementary Table 8)[11,13,24,25]. The data obtained by other methods (e.g., Fourier transform infrared spectrophotometer or Raman spectroscopy) are not included due to the large discrepancies with the visual identification method (Supplementary Discussion)[26]. Mass concentrations are used to compare with the observed surface concentrations as plastic discharge inventories are also in a mass unit, while number concentrations are transformed to mass concentrations as they are prone to larger uncertainties due to the fragmentation processes (Supplementary Discussion)[11]. We employ a three-dimensional variational method to derive an optimally estimated

global plastic emission to oceans based on prior emission estimates and seawater plastic observations. A super ensemble containing 156 (=52 × 3) members is constructed by sampling the uncertainties of modeled transport and transformation processes ($n = 52$) and the existing emission inventories ($n = 3$). The optimization process is repeated separately with each sampled member to generate an ensemble of optimally estimated emissions, which represents the uncertainties of our estimations (see the "Methods" section).

## Results and discussion
### Ocean surface plastics
We choose three emission scenarios to represent the range of uncertainties originating from emission inventories in the literature: (1) High: riverine emissions as suggested by Lebreton et al.[5]; (2) Middle: riverine emissions by Mai et al.[7]; and (3) Low: riverine emissions by Weiss et al.[9]. The Lebreton and Mai inventories are for total plastics, including both microplastics (diameter <5 mm) and macroplastics (diameter > 5 mm), while the Weiss inventory only includes microplastics. In a sensitivity analysis, we transfer the microplastic emissions by the Weiss inventory to total plastics according to the mass fraction of microplastics in river samples [41–48% according to Weiss et al.[9]], but the optimized emissions are not significantly influenced. For each scenario, we also consider the direct emission from coastal zones following Jambeck et al.[4] and marine sources from shipping and fishing activities with the latter two scaled by the riverine emissions[27], resulting in global emissions of 7.1, 0.68, and 0.031 million Mt yr$^{-1}$ for the three scenarios, respectively (see the "Methods" section). According to Wilcox et al., the historical trends of emissions from 1950 to 2018 are assumed to follow the trend of accumulated global plastic production (Supplementary Fig. 1)[28].

The modeling results from the three scenarios reveal large spatial variability for plastic distributions in the surface ocean and successfully replicate the development of "Garbage Patches" in the subtropical ocean gyres in both hemispheres (Fig. 2a–c). In these regions with anticyclonic wind stress, Ekman transport, and direct windage effects result in a convergence zone that concentrates buoyant plastics[27]. Taking the Middle scenario as an example, the "test case" model (Table 1; see the "Methods" section for more details) simulates that the concentrations of plastics accumulating in these regions range between 0.2 and 4.4 kg km$^{-2}$ while the observed range is <0.01–16 kg km$^{-2}$. Compared with the northern hemisphere, accumulation zones in the South Pacific Ocean and South Atlantic Ocean present lower modeled concentrations (0.2–1.4 kg km$^{-2}$ vs. <0.01–2.3 kg km$^{-2}$ in observation) due to the lower terrestrial discharge. The model allows us to identify additional zones of plastic accumulation. Heavy discharge from Asia results in 0.2–6.1 kg km$^{-2}$ in the western subtropical Pacific Ocean (especially the coastal regions) and the highest observed concentration is 14 kg km$^{-2}$. Simulated concentrations near the discharge points along the coastlines of South Asia and Europe are also relatively high (up to 5.1 vs. 3.1 kg km$^{-2}$ in observation). Relatively higher concentrations are modeled in the Atlantic sector of the Arctic Ocean (0.24 kg km$^{-2}$) than in other remote areas due to intense fishing and ship traffic[29].

The three scenarios simulate drastically different concentration levels in the surface ocean, resulting from the large differences in emissions. The High, Middle, and Low scenarios generate a plastic mass of 579, 55, and 2.7 thousand Mt in the global surface ocean, respectively, spanning 2.3 OMs. Figure 2 also contrasts the model results with observed surface plastic concentrations. We find that the Middle scenario agrees with the available observations the best, with 51% (49%) of the data points above (below) the 1:1 line (total $n = 764$ after averaging the observations within the same 2° × 2.5° model grid), whereas the High and Low scenarios simulate 86% (14%) and 6% (94%) above (below) the 1:1 line, respectively. The High and Middle scenarios

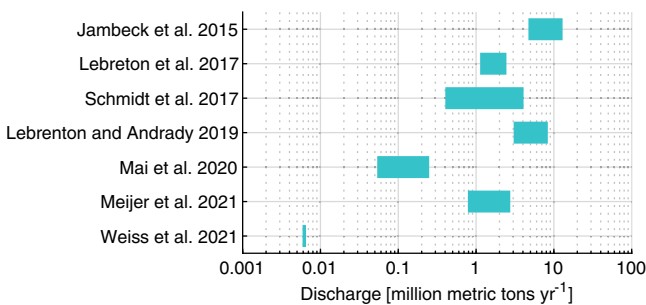

**Fig. 1 | Estimated global total plastic waste discharge (million metric tons year$^{-1}$) from land to ocean.** Note that the x-axis is logarithm-based. Jambeck et al.[4] consider the emissions from the coastal zones, while Lebreton et al.[5], Schmidt et al.[83], Mai et al.[7], Meijer et al.[8], and Weiss et al.[9] consider only riverine discharge, and Lebreton and Andrady[84] include both riverine and coastal zone sources.

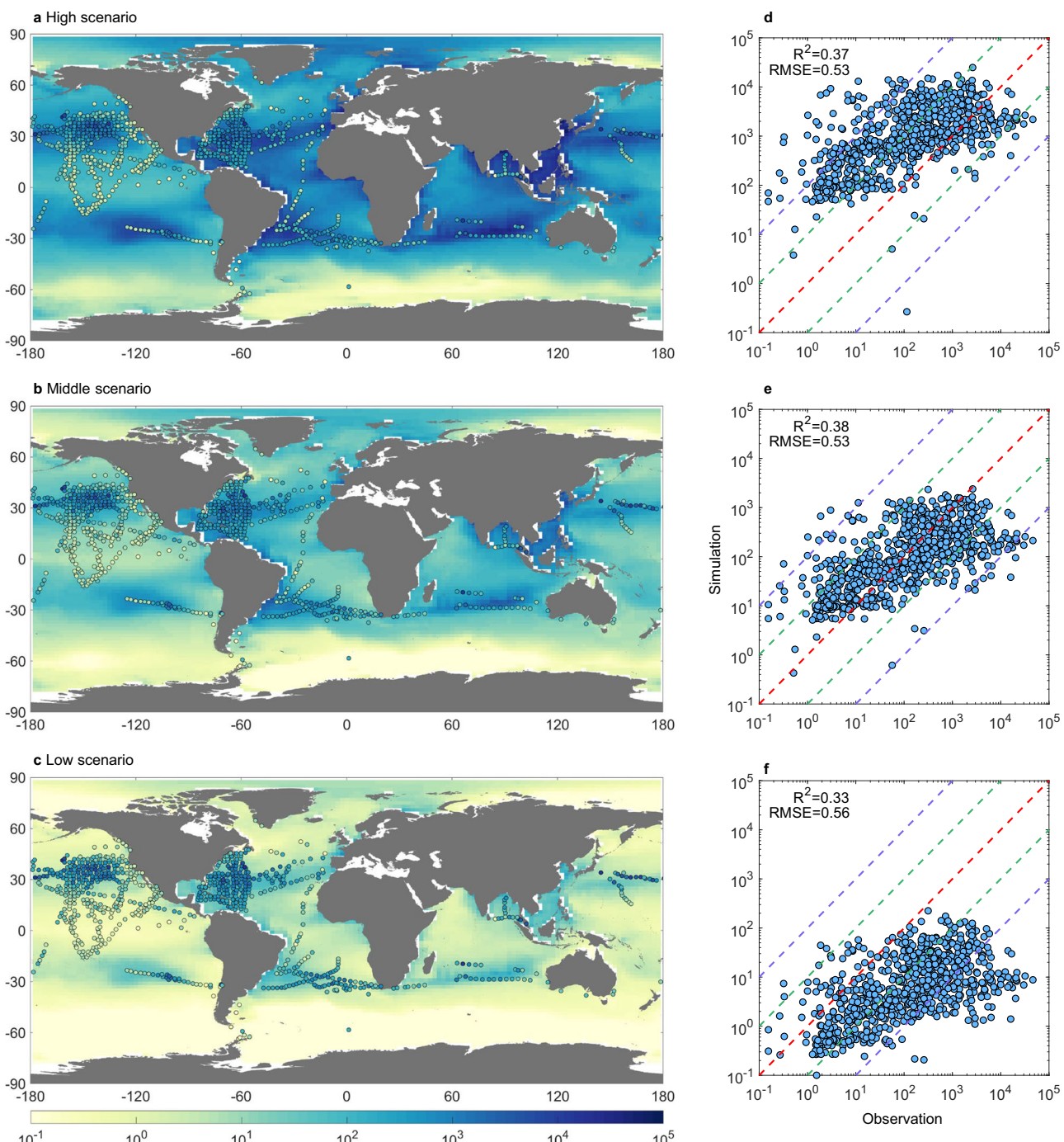

**Fig. 2 | Comparison between the modeled and observed total surface ocean plastic mass concentrations. a, d** High emission scenario. **b, e** Middle emission scenario. **c, f** Low emission scenario. In panels **a–c**, the background colors are the modeled annual mean for the year 2018, while the circles are observations. In panels **d–f**, the dashed lines are 100:1, 10:1, 1:1, 1:10, and 1:100. All the concentrations and the calculation of $R^2$ and RMSE are in a base 10 logarithmic scale with a unit of g km$^{-2}$. Sources of observation data are listed in Supplementary Table 8.

have similar $r^2$ values (coefficient of determination for a linear regression model, 0.37 and 0.38), slightly higher than the Low scenario (0.33). All the emission scenarios tend to underestimate the highest observed concentrations in the gyre centers, likely caused by numerical diffusion due to the coarse model resolution[17]. Most previous models do not consider real plastic emissions and thus are not comparable with observations. Some models also have issues in reproducing the observations in the low concentration ranges or simulating inaccurate ocean locations for the highest plastic concentration patches[12].

## Model uncertainties

The model has large uncertainties due to the relatively coarse resolution and simplified ocean physics. The effects of some small-scale processes such as anticyclonic and cyclonic eddies are not included in the model. These processes are quite complicated and are found to contribute inversely in different studies[30,31]. In addition, the present understanding of the physical and biogeochemical processes of plastics in the ocean is incomplete. There is indeed a trade-off between a finer resolution and a larger number of sensitivity experiments. We set many model scenarios to test the sensitivity of model results to the

**Table 1 | Model parameter sensitivity analyzed in this study**

| Model parameter | | Scenario |
|---|---|---|
| Test case | | The beaching rate is 10% day$^{-1}$. |
| | | The rate of fragmentation and abrasion is 10% yr$^{-1}$. |
| | | The effect of biofouling is not modified. |
| | | The effect of sedimentation is not modified. |
| | | The ocean source accounts for 20% of the total discharge. |
| Beaching rate[a] | 1%/day | The beaching rate is 1% day$^{-1}$. |
| | 5%/day | The beaching rate is 5% day$^{-1}$. |
| | 25%/day | The beaching rate is 25% day$^{-1}$. |
| | Constant | The beaching rate is constant on the global coasts, regardless of sandy beach length. |
| Fragmentation rate | 1%/yr | The rate of fragmentation and abrasion is 1% yr$^{-1}$. |
| | 10%/mon | The rate of fragmentation and abrasion is 10% mon$^{-1}$. |
| | Depending on size | The rate of fragmentation and abrasion for microplastic is 10% mon$^{-1}$ while the rate for macroplastic is 10% yr$^{-1}$. |
| | Depending on type | The rate of fragmentation and abrasion for PE is 10% yr$^{-1}$, PP and PVC are 1% yr$^{-1}$, and others are 10% mon$^{-1}$. |
| Biofouling rate | Low | The effect of biofouling is reduced by a factor of 0.1. |
| | High | The effect of biofouling is magnified by a factor of 10. |
| Sedimentation rate | Low | The effect of sedimentation is reduced by a factor of 0.1. |
| | High | The effect of sedimentation is amplified by a factor of 10. |
| Marine sources | Low | There is no ocean-based discharge. |
| | High | The ocean source accounts for 40% of the total discharge. |

[a]The beaching rate here is not corrected by the portion of sandy beaches (see the "Methods" section).

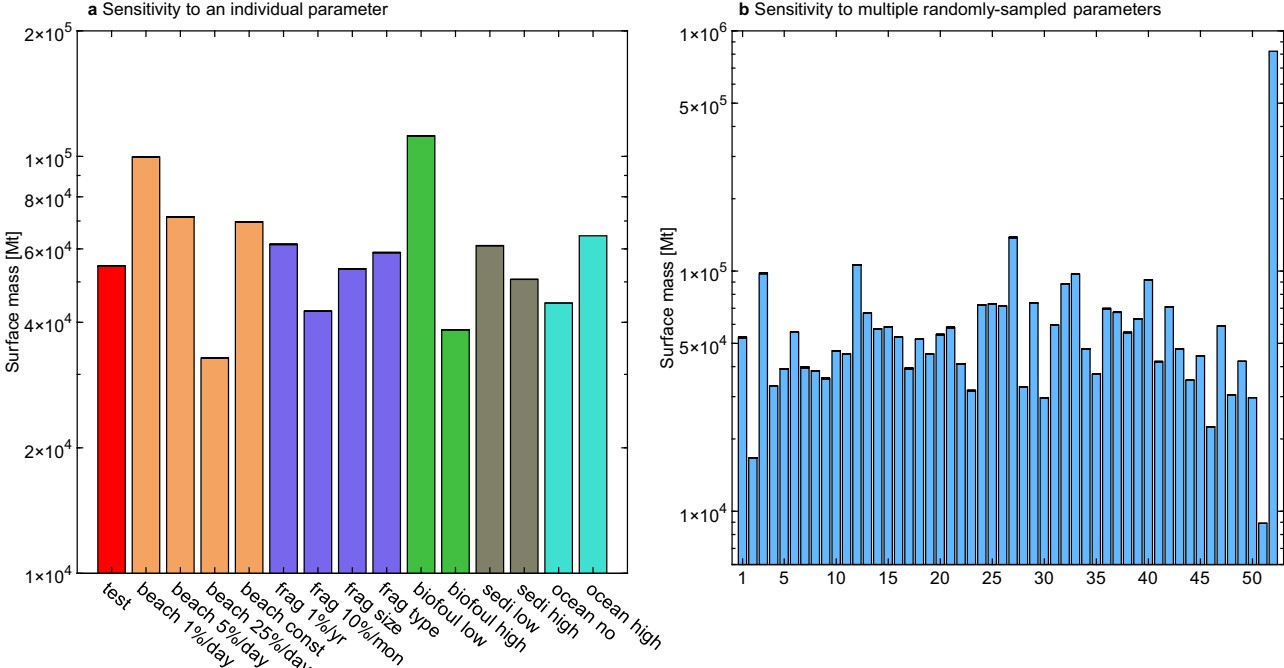

**Fig. 3 | Sensitivity of the modeled total plastic mass in the surface ocean (Mt, metric tons) to different parameters. a** Sensitivity to an individual parameter. **b** Sensitivity to multiple randomly-sampled parameters. Note the y-axis is on a logarithmic scale. The model scenarios include varying beaching, fragmentation, biofouling, and sedimentation rates. Additional members consider uniform beaching rates for all beaches, fragmentation rates depending on particle size and chemical composition, and low and high marine sources (Table 1).

model parameters (Table 1 and Fig. 3a), which allows us to assess the model uncertainty and compare it against that of plastic emissions. These model scenarios have different fragmentation/abrasion, beaching, biofouling, and sedimentation rates with values ranging as reported in the literature and are driven by the same emission scenario (the Middle one). Also, our model focuses on the global scale so the impact of small-scale processes seems to be of second-order importance.

In the previous studies, the fragmentation/abrasion rates are a function of environmental factors including light, temperature,

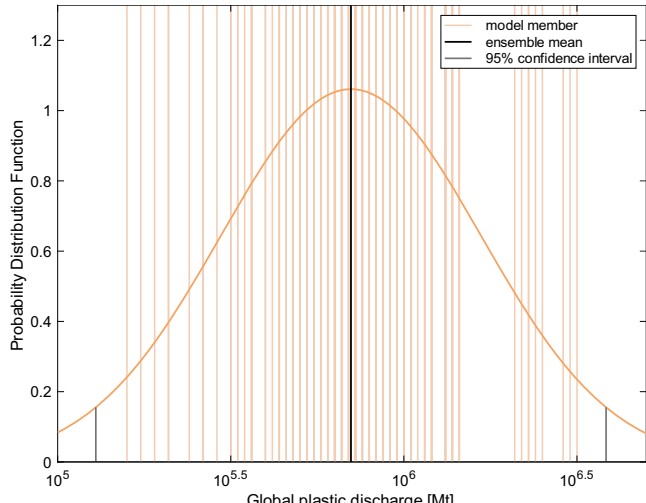

**Fig. 4 | Optimized global plastic discharge (Mt, metric tons).** The estimation is derived from a three-dimensional variational method powered by a super ensemble of model parameters and emission inventories. The thicker black line represents the ensemble mean with thinner orange ones for each member. The curve is the probability distribution function (pdf) of the optimized global plastic emissions, which is assumed to be a log-normal distribution based on the simulated surface plastic mass (Supplementary Fig. 2). The short gray vertical lines mark the 95% confidence interval. The minimum and maximum discharges derived from models #52 and #51, which are not shown in the figure, are $10^{2.9}$ and $10^{7.0}$ Mt, respectively.

oxygen, and plankton biomass[32]. Limited experimental data show that these rates could vary for more than an OM among plastic types and environmental conditions (see the "Methods" section). We find that perturbing this rate by a factor of 10 causes a 22% change in the total plastic mass on the surface ocean as the deposition and sedimentation rates are functions of particle sizes (Fig. 3a). The modeled fraction of uncatchable plastics with diameters less than 0.33 mm that is the standard mesh size of neuston trawls, varies between 0.18% and 18% in the surface ocean. Plastics below this size could not be caught by regular trawls and cause an underestimation of surface plastic concentrations in the observation dataset. Our model thus could appraise such underestimates. The model also suggests that more uncatchable particles are generated with stronger fragmentation/abrasion. In another scenario, we specifically increase the fragmentation/abrasion rate of microplastics by 10 times (but keep that of macroplastics unchanged), as small plastic degrades faster than large ones due to a higher surface-area-to-volume ratio[13]. The fraction of uncatchable plastics in the surface ocean also increase to 5.1%.

Another major source of uncertainty is the biofouling of plastics by marine algae, which is the main driver of the vertical transport of positive-buoyant plastics[33]. The biofouling rate is determined by algae attachment, growth, respiration, and mortality[34]. These processes are considered as the transformation routes among three types of plastics: floating, neutral, and sinking with increasing degrees of biofouling and density in this model (see the "Methods" section). The rates of different processes in our model follow those of Kooi et al., which are estimated based on literature data for an average marine algal species[33]. These parameters, however, often vary by 0.5–1 OM among different algal species[35]. Experimental data also indicate the surface longevity of plastic particles of different sizes varies by a factor of 4 (17–66 days) due to biofouling[36]. Perturbing the biofouling rate by an OM results in a 0.47 OM change in total plastic mass in the surface ocean (Fig. 3a), smaller than the results of Kvale et al.[37]. The smaller range of change is probably attributed to the highly simplified removal effects of zooplankton and marine snow which are represented by the plankton biomass in our model.

The beaching/sedimentation rates are an important source of uncertainty, as beaches and sediments are major reservoirs for marine plastics[18,38]. The rates of both beaching/sedimentation, resuspension, and the ultimate residence time of plastics on beaches/sediments depend on the characteristics of plastics, the morphological features of beaches/seafloor, wind, circulation, and wave conditions[18,39–41]. We perturb the beaching rate from 1% to 25% day$^{-1}$, which represents two extremes of the behavior of plastic particles, i.e., nearly no beaching and fast capture by beaches, as suggested by previous empirical studies[40,42–44]. The model suggests that the beached plastics account for 8.1–63% of total discharge in these scenarios (Fig. 3a). A similar level of uncertainty is found for the plastic sedimentation rate, which could vary as high as 2 OMs as suggested by Kane et al.[45].

Plastic waste from shipping and fishing activities is a non-negligible source of ocean plastics[46–48]. For example, the increasing number of plastic drink bottles found in the ocean indicates that large amounts of plastic waste originate from ships[48]. However, attempts to estimate the magnitude of the marine source are scarce and only a few studies assume that the marine source accounts for 20% of the total plastic discharge[27,45]. Not considering the marine source results in an 18% reduction in the total plastic mass on the surface ocean compared with the test case scenario. The surface plastic mass increases by 30% when the fraction of the marine source increases from 20% to 40% (Fig. 3a).

## Model ensemble and optimal emission

We evaluate the overall uncertainty associated with these model parameters by a Monte Carlo approach as multiple parameters can vary in a wide range simultaneously and could have significant interactions with each other (Fig. 3b and Supplementary Fig. 2). Driven by the Middle emission scenario, ensemble of 50 models is generated with multiple model parameters randomly sampled from their literature-reported ranges (Supplementary Table 7 and Supplementary Fig. 3). We find that different ensemble members span about an OM in predicting the total surface ocean plastic mass. This variability range is larger than that of the individual parameters as discussed above (about 0.5 OM). Indeed, the lowest (#2) and highest (#27) members both have more than one parameter with relatively extreme values (Supplementary Table 7). We also include another two models (#51 and #52) that sample the lowest/highest parameters to represent the largest possible collective variability and the surface mass range. The total surface ocean plastic mass predicted by these two models varies by about 2 OMs. Overall, we find that the variability caused by different emission inventories (spanning 2.3 OMs) far surpasses that of model parameters (spanning 1–2 OMs), indicating that the model can be potentially effective in constraining the magnitude of emissions.

Figure 4 shows the probability distribution of the optimized global emissions of marine plastics by employing a super ensemble three-dimensional variational method (see the "Methods" section). The method optimizes the global plastic emission by minimizing a cost function that measures the deviation from the prior emission inventory and the observed surface ocean plastic abundance (Eq. (24)). Such an optimization process is repeated for each member of the super ensemble, which is consisted of the same 52 model members as above but is driven by all three emission inventories. This results in a total of 156 (=3 × 52) optimal estimates for the global plastic emissions, and the mean (taken as the best estimate) is 0.70 million Mt yr$^{-1}$ with a 95% confidence interval of 0.13–3.8 million Mt yr$^{-1}$, spanning about 1.5 OMs.

We note that the variability among different emission inventories (1.4 OMs) far surpasses that of the observed ocean plastic abundance data (mean 0.38 OM). The cost function thus puts much more weight on the deviation from observed ocean plastic abundance data than from the prior emission inventories. In another word, the optimized emissions are mainly achieved by minimizing the model- and observation-derived surface ocean plastic concentrations, while the

priori emission estimates have a relatively small influence. However, this does not mean the prior emission estimate is useless, as the optimized emissions follow the same spatial and temporal distribution, chemical composition, source contribution, and size distribution of the plastic emissions from the prior emission inventory. The importance of the prior emission estimate would increase as the knowledge of emission inventory is fortified and the prior emission uncertainty better constrained.

The uncertainty of our optimized estimate for plastic emission is relatively high (-1.5 OM). First, it is contributed by the variability of the model parameters represented by the model ensemble. The modeled ocean concentrations driven by the best-estimate emissions have a mean absolute error of 0.68 OM compared to observations, and 77% of the points have an error less than an OM (Supplementary Fig. 4), which may represent additional model uncertainty not characterized by our model ensemble. The evolving understanding of the environmental processes for marine plastics will narrow down the uncertainty range of our optimized estimate. Second, the top-down approach used in this study limits us to discern emissions from riverine discharge, direct emissions from coastal zones, or oceanic sources, as well as their spatial and temporal patterns. The continuously accumulating observations may also allow us to constrain source-specific and/or regional emissions to individual ocean basins in the future. Third, our results are dependent on available observed surface plastic abundance datasets during 2000–2015 by "sporadically" cruise studies that are often not repeated and mostly located in the centers of gyres. More observed data that cover the global ocean (e.g., over the southern hemisphere) and a longer time period can increase the reliability of our optimal estimation. Fourth, the observations of plastic abundances in other compartments, e.g., water columns and beaches, are rather limited to evaluate the model. Our model captures the observed vertical trend, indicating a reasonable representation of the fraction of plastic mass in the surface ocean (Supplementary Fig. 5). The model ensemble yields a wide range of fractions of the beached plastics to the total discharge (Supplementary Fig. 7), which brackets the potentially highly variable beached fraction in the real ocean. The high end of the range, which is less likely to occur in the ensemble, is possible to be higher than the observations[19,49,50]. We thus suggest that a combination of both bottom-up and top-down approaches, such as developing more accurate emission inventories, obtaining more data for abundances of plastics in seawater and other compartments, and measuring more accurate model parameters, would be the future research directions.

As an independent alternative, the present study illustrates a top-down approach to constrain global ocean plastic emissions by taking advantage of seawater measurements. We substantially narrow down the uncertainty of the plastic emissions to the global ocean from 2–3 OMs to -1.5 OM. A more accurate estimate of the global ocean emissions helps to understand the human perturbation on the marine environment and the fate of plastics in the ocean[9]. Combined with plastic waste management and MPW generation information, more effective mitigation strategies and measures could also be designed and implemented.

## Methods
### General description of model
We simulate the fate and transport of plastics by the Nanjing University Marine Plastic (NJU-MP) model based on the MITgcm model framework[51,52]. This model is Euler-based and simulates the emission, transport, diffusion, sinking, and transformation (including biofouling, fragmentation, and abrasion) of plastics in each model grid cell. The model has a resolution of 2° × 2.5° horizontally with 22 vertical levels, and a time step of 4 h. The ocean circulation data are from the Integrated Global Systems Model (IGSM)[53]. The ocean boundary layer physics is modeled based on Large et al.[54], and the effects of mesoscale

eddies on isopycnal mixing are parameterized following Gent and McWilliams[55]. We run the model from 1950 to 2018. The model has a relatively coarse resolution to resolve the currents over coastal regions and western boundary currents such as Kuroshio and Gulf Stream but performs better over the open ocean[56]. The lower computational costs compared to high-resolution models also allow us to perform long-term and multiple-scenario runs. We test different time steps and the results keep relatively robust due to the low stiffness in simulating these processes. The model includes five categories of different chemical compositions and the density of each category is pre-determined: polyethylene (PE, 950 kg m$^{-3}$), polypropylene (PP, 900 kg m$^{-3}$), polyvinyl chloride (PVC, 1410 kg m$^{-3}$), polystyrene (PS, 1050 kg m$^{-3}$), and acrylonitrile butadiene styrene (ABS, 1050 kg m$^{-3}$). Each category has six size bins with bounds equally distributed on a log scale: four for microplastics: <0.0781, 0.0781–0.3125, 0.3125–1.25, and 1.25–5 mm, and two for macroplastics: 5–50 and >50 mm. The plastics' density is increased when biofouled and we consider three states for PE and PP: floating, neutral, and sinking (no floating and neutral states for PVC, PS, and ABS as they are originally heavier than or as heavy as seawater). There are a total of 54 plastic tracers in the model (PE and PP each have 18 tracers while the other three each have 6 tracers), representing the common chemical compositions, size ranges, and biofouling states of plastics in the marine environment.

We develop a universal framework for the transport of these tracers in the global ocean. Tracking the three-dimensional motion of plastic particles is distinct from estimating other trace components in ocean models since plastic particles have non-negligible volumes and different densities from seawater. Obtaining closed expressions describing the hydrodynamic forces experienced by rigid particles embedded in various flows has been a subject of active research for a long time[57]. Equations could be selected and simplified, based on facts, to help the simulation. Most of the time in the global ocean, light particles (PP and PE) float and drift in quasi-two-dimensional motion relative to the sea surface. Light particles can be biofouled and sink to the subsurface ocean, where they can rise again after defouling. Other particles that have a higher density than the seawater (e.g., PVC), would instantly sink after being dumped. Sinking or rising particles make an approximate one-dimensional motion relative to the water column and the velocity depends on their densities and diameters. Thus, in our model, the motion of plastic particles is resolved into advection and mixing (three-dimensional), sinking/rising (one-dimensional), and drifting (two-dimensional). The above processes are computed separately in the model (aka operator splitting) though they occur simultaneously in the real ocean. We calculate particles' sinking, rising, and drifting velocities by fluid dynamics and empirical equations. Without loss of generality, plastic particles in our model are treated as smooth rigid spheres. These processes are elaborated on in the sections below.

### Plastic sources
We use the riverine plastic emission inventories from Lebreton et al.[5], Mai et al.[7], and Weiss et al.[9], and are referred to as the High, Middle, and Low scenarios, respectively. The Lebreton inventory uses data from a global compilation of solid waste for 105 countries by the World Bank[58]. It considers the seasonality, spatial variability, and size distribution of local sources. This inventory estimates plastic discharge based on waste management, population density, and hydrological information, and uses MPW as a proxy to predict the discharge from rivers without observation data. They also divided the total plastic discharge into micro- and macroplastics according to the ratio of them in sampled rivers. Mai et al.[7] and Weiss et al.[9] compiled available riverine discharge data and developed regression models between the discharge of individual rivers and the characteristics of the rivers (e.g., water discharge) and the countries in the corresponding watersheds (e.g., population density, solid waste generation per capita, the fraction of plastics in solid waste, and waste treatment rate). A key feature

of the Mai inventory is a prediction model for the waste treatment rate by Human Development Index (HDI) based on an updated database by World Bank[59]. Weiss et al. use a lower representative mass for microplastics in rivers and are limited to data collected by plankton net sampling[9]. For these two inventories, we fill the data gap in the spatial, seasonal, and size (i.e., micro vs. macroplastics, refer to Supplementary Table 1 for its mapping to our modeled size bins) distributions of emissions following the Lebreton inventory.

The coastal plastic emission inventory follows the framework of Jambeck et al.[4] which uses MPW as proxy data to estimate plastic discharge but has a lower estimation by Chassignet et al.[60]. Similar to the Lebreton inventory, it also relies on solid waste data from the World Bank[58]. There is double counting for emissions from coastal areas that are also located near river mouths, but the overall contribution is negligible[60]. This Jambeck/Chassignet inventory is directly used for coastal emissions when the Lebreton inventory is chosen for rivers. When the other two lower riverine inventories are chosen, the above coastal inventory is also scaled down to keep the ratio between the riverine and coastal emissions constant. In addition to the riverine and coastal plastic emissions, we consider direct ocean emissions from marine activities such as shipping and fishing, which are assumed to account for 25% of terrestrial discharge for all three emissions scenarios[27,45]. The spatial pattern of the discharge from marine activities is allocated according to the global footprint of fisheries and shipping tracks[61,62].

Cumulative plastic production is used as a proxy for the historical trend of plastic discharge from the terrestrial environment during 1950–2018[6,28]. The historical trend of plastic discharge from each continent is assumed to follow that of per-capita gross domestic product (GDP)[63]. The discharge of each type of plastic is allocated by its proportion to global consumption in 2013 due to the lack of consumption data for the whole period (Supplementary Table 3)[64]. The historical trend of marine emissions follows that of fishery and shipping activities[62]. The emissions from shipping activities are expected to decrease substantially since intentional disposal was banned when the MARPOL Convention went into effect in the 1980s[65]. There might still be unintentional disposal due to container loss and accidents, but we assume that this source is zero due to a lack of data. The resulted global plastic emissions to the ocean for 2018 are thus estimated as 7.1, 0.68, and 0.031 million Mt for the High, Middle, and Low scenarios, respectively. The corresponding historical total emissions are 161, 15, and 0.70 million Mt, respectively (Supplementary Fig. 1).

## Sinking and rising

The plastic densities that depend on their polymer types, together with their diameters and shapes, and seawater state, determine their sinking/rising velocities. In a steady-state, three balanced vertical forces act on the particles:

$$F_g = V_p \rho_p g \tag{1}$$

$$F_b = -V_s \rho_s g \tag{2}$$

$$F_D = -\frac{1}{2} C_D(\mathrm{Re}_s) A_p \rho_s \frac{(w - w_s)^3}{|w - w_s|} \tag{3}$$

$$F_D + F_g + F_b = 0 \tag{4}$$

where $F_D$ is the vertical dragging force of the seawater, $F_g$ is gravity, and $F_b$ is buoyancy. $V_p$ is the volume of the particle, while $V_s$ is the volume of the particle that is submerged in seawater ($V_p = V_s$ in this case, but $V_p > V_s$ for floating particles with zero sinking/rising velocity relative to the seawater, e.g., clean PP and PE). $C_D$ is the coefficient of dragging,

which is a function of the Reynolds number (Re) of a certain motion of a fluid. $A_p$ is the horizontal sectional area of a particle, $\rho_s$ is the density of seawater, $\rho_p$ is the density of a particle, $w$ is the vertical velocity of the particle, $w_s$ is the velocity of seawater, and $g$ is the gravity acceleration.

Based on Eqs. (1)–(4), we get:

$$(w - w_s)^2 = \frac{4|g| d(\rho_p - \rho_s)}{3 C_D(\mathrm{Re}_s) \rho_s} \tag{5}$$

where $d$ is Stokes diameter of a particle, and $C_D$ is calculated as[66]

$$C_D(\mathrm{Re}) = \begin{cases} 24\mathrm{Re}^{-1} & \mathrm{Re} \le 0.3 \\ 18.5\mathrm{Re}^{-0.6} & 0.3 < \mathrm{Re} \le 1000 \\ 0.44 & 1000 < \mathrm{Re} \le 20{,}000 \end{cases} \tag{6}$$

$\mathrm{Re}_s$ is the Re of seawater and is calculated as:

$$\mathrm{Re}_s = \frac{d \rho_s |w - w_s|}{\mu_s} \tag{7}$$

where $\mu_s$ is the dynamic viscosity of seawater. By substituting Eqs. (6) and (7) into Eq. (5), with a few techniques (Supplementary Methods), we can solve $w$[67]. In this way, we get the rising or sinking speed of the particles, by which we simulate the vertical transport of the plastic particles in the seawater columns (Supplementary Table 4).

## Drifting

Plastic particles floating on the sea surface are subject to wind forces, which are commonly referred to as leeway drift, or windage[68]. We consider the windage of all the plastic size categories, including both microplastics and macroplastics[69]. The motion of a drifting particle in balance, which is affected by five forces (gravity, buoyancy, seawater stress, horizontal wind stress, and Coriolis force), is described by Eqs. (1), (2), (8), (9), and (10), respectively.

$$F_s = -\frac{1}{2} C_D(\mathrm{Re}_s) A_s \rho_s \frac{(u - u_s)^3}{|u - u_s|} \tag{8}$$

$$F_a = -\frac{1}{2} C_D(\mathrm{Re}_a) A_a \rho_a \frac{(u - u_a)^3}{|u - u_a|} \tag{9}$$

$$F_c = V_p \rho_p f_C u \tag{10}$$

$$\mathrm{Re}_a = \frac{d \rho_a |u - u_a|}{\mu_a} \tag{11}$$

where subscript p denotes plastic particle, a denotes air (or wind), and s denotes seawater. $F_s$ and $F_a$ are the horizontal dragging force by seawater and air, respectively. $u$, $u_s$, and $u_a$ are the velocity of plastic particles, seawater, and wind, respectively. $A_a$ and $A_s$ are the vertical sectional areas of particles exposed to the air and seawater, respectively. $\rho_a$ and $\mathrm{Re}_a$ are the density and the Reynolds number of air, respectively. $\mu_a$ is the dynamic viscosity of air. $f_C$ is the Coriolis parameter.

We assume the vertical forces act on a particle floating on the ocean surface to reach a balance:

$$F_g + F_b = 0 \tag{12}$$

with which we can solve the $V_s$ (in this case, $V_s < V_p$) and subsequently $A_a$ and $A_s$ based on geometry (Supplementary Table 5). $C_D(\mathrm{Re}_a)$ and

$C_D(\text{Re}_s)$ are calculated as functions of Reynolds number of air (Eq. (11)) and seawater (Eq. (7)), respectively, based on Eq. (6).

We solve $u$ by assuming the horizontal forces $F_a$ and $F_s$ reach a balance while $F_c$ is neglected due to a much smaller magnitude:

$$F_a + F_s = 0 \qquad (13)$$

By substituting Eqs. (8) and (9) into Eq. (13), we solve $u$ numerically by a gradient descent method. This leads to a constant drifting speed $u$ given $u_a$ and $u_s$. The $u_a$ is from NCEP/NCAR reanalysis[70]. The random walk of plastic particles caused by oceanic eddy turbulence with a scale that is smaller than the grid size is simulated as an isopycnal diffusion process in the model, which follows the Laplacian operator with the mixing coefficients given in Dutkiewicz et al.[35].

Stokes drift, a near-surface velocity induced by ocean wind-generated gravity waves, can contribute to the near-surface transport of mass including plastic particles[71]. For example, it changes the locations of convergence zones for plastic mainly caused by Ekman currents, resulting in a westward entrainment in the north of the convergence zone in the Indian Ocean[72]. We consider the Stokes drift for plastic tracers on the surface ocean in the model. The surface Stokes drift estimates are from the GlobCurrent[73] and the annual mean from 1990 to 2015 is applied for the whole simulation period.

## Fragmentation/abrasion

Fragmentation represents the process during which large plastic particles break up into smaller ones. Abrasion refers to the process in which tiny plastics peel off from the surface of larger ones, usually caused by mechanical shearing[74]. The rate of fragmentation in seawater has limited data but varies drastically depending on the type and shape of plastics, as well as the environmental factors[75,76]. For example, in a laboratory experiment, seawater PE, PP, and PS lost ≤1% of their weight per year while the ratio of loss was higher (3–27%) for other polymers such as polyurethane and polyester[76]. In another study, PE and PP were found to have higher fragmentation rates than 1% yr⁻¹. They lost 0.39–1.02%, with averages of 0.45% and 0.39%, respectively, of their masses per month[75]. Considering the various fragmentation rates of different types of plastics, the total fragmentation and abrasion rate $R$ (% yr⁻¹) in seawater for all plastics is assumed to be 3–30% yr⁻¹, following a log-normal distribution. Also, 10% yr⁻¹ of the total $R$ (i.e., 0.3–3% yr⁻¹) is allocated as the abrasion rate[77]. The rate in the surface ocean ($R_{surf}$) is increased proportionally to the downward shortwave solar radiation ($q$, also a proxy for temperature) in the surface ocean if the plastic particles are not biofouled, reflecting the dependence of weathering rate on sunlight revealed by controlled experiments[78,79]:

$$R_{surf} = \frac{R}{175} \cdot q + R \qquad (14)$$

The surface downward shortwave radiation is taken from the CMIP5 project, which causes a factor of approximately three between the polar and tropical regions.

## Beaching

A plastic particle has a chance to be 'beached' (i.e., deposited onto beaches) when it arrives at a beach-adjacent cell. The chance or "beaching rate" is geographically diverse depending on the coastal morphological features, wind, and wave conditions. Previous studies suggest that beached plastic can be eroded back into the ocean, and bi-directional exchanges occur between the beach and coastal seawater[40]. Atwood et al. found that <10–94% of released microplastics were beached and the majority of beaching occurred within the first three days, which is about 3.3–31% per day[42]. Ocean drifter studies revealed that the timescales of the beaching and resuspension processes ranged from 3 to 4 weeks under different

conditions[40,43,44]. These timescales can be transferred to beaching rates between 100% per 21 days and 100% per 28 days, i.e., 3.6–4.8% day⁻¹. We simulate the beaching process by a net beaching rate (i.e., beaching−resuspension) in this study due to the large uncertainty of the two processes and the coarse resolution of our model. We assume that the plastics in grid cells immediately adjacent to sandy beaches are partially removed from the seawater and the mass of beached plastics is proportional to the length of the beach in the cell. We assume a net beaching rate in a cell as 2.5–18% day⁻¹. Their dynamics of beaching and resuspension are highly variably on local scales. Not all sandy beaches have a net accumulation of plastic debris, and not vice versa. We thus consider the fraction of sandy beaches as a proxy for the coastal morphological features. So, the effective beaching rate is corrected by the portion of sandy beaches in a cell, which results in a rate between 0.15% and 1.1% day⁻¹. The global sandy beaches dataset is from Luijendijk et al.[80]. The global estimated percentage of sandy shorelines varies from 10% to 75% due to the lack of reliable global-scale assessment of occurrence or rates of sandy shoreline change.

## Biofouling and defouling

Biofouling of light plastic types (PE and PP) is modeled following Kooi et al. which is based on the Lagrangian perspective but adjusted for our Eulerian-based framework[33]. Kooi et al. directly tracked the growth and respiration of plankton/microbe on the particles but such a process is hard to be replicated in an Euler model because we do not know the history/trajectories of these particles. This process has an impact on the density of plastic particles and is simulated with three types of tracers in our model: floating, neutral, and sinking. Floating plastics have no biomass attached and stay floating on the sea surface with the same density as the original plastic materials. Neutral plastics are assumed to be neutrally buoyant and suspended in water columns. Sinking plastics are heavier than seawater and sink into the subsurface ocean. The biomass of plankton in global oceans is used as a proxy to scale the overall biofouling potential from microbes, phytoplankton, zooplankton, marine snow, and the ingestion and inclusion in feces. The plankton biomass data is taken from the MITgcm Darwin ecosystem model[35]. Using plankton as a proxy might introduce uncertainty as the community structure varies drastically in different ocean biotic provinces. The Darwin model also does not consider the vertical migrations of zooplankton. But it is relatively robust as it is constrained by satellite remote sensing data[35]. Especially, the plankton distribution data suggest stronger biofouling in productive coastal waters than in the open ocean, consistent with empirical studies[14].

The volume of biomass on plastic particles ($V_{bf}$) depends on the algae volume $V_a$, the number density of attached algae per unit area $A$ (# m⁻²), and the surface area of the plastic particle $\theta_p$[33]:

$$\frac{dV_{bf}}{dt} = V_a \theta_p \frac{dA}{dt} + V_a A \frac{d\theta_p}{dt} \qquad (15)$$

where:

$$\frac{dA}{dt} = \frac{\beta_a A_a}{\theta_p} - m_a A \qquad (16)$$

where $\beta_a$ is the encounter kernel rate (m³ s⁻¹), $A_a$ is the ambient algae concentration (# m⁻³), and $m_a$ is the mortality rate (s⁻¹). The encounter kernel rate $\beta_a$ is the sum of Brownian motion and advective shear collision frequencies (m³ s⁻¹):

$$\beta_a = \beta_{brownian} + \beta_{shear} \qquad (17)$$

where:

$$\beta_{\text{brownian}} = 4\pi(D_{\text{p}} + D_{\text{a}})(r_{\text{p}} + r_{\text{a}}) \qquad (18)$$

$$\beta_{\text{shear}} = 1.3\gamma(r_{\text{p}} + r_{\text{a}})^3 \qquad (19)$$

$$D_{\text{p}} = \frac{k(T + 273.16)}{6\pi\mu_{\text{sw}}r_{\text{p}}} \qquad (20)$$

$$D_{\text{a}} = \frac{k(T + 273.16)}{6\pi\mu_{\text{sw}}r_{\text{a}}} \qquad (21)$$

where $D_{\text{p}}$ and $D_{\text{a}}$ are the diffusivity of plastics and the individual algae cells ($m^2\,s^{-1}$), respectively, $r_{\text{p}}$ and $r_{\text{a}}$ are the radius of plastics and the individual algae cells, respectively, $\gamma$ is the shear rate ($s^{-1}$), $k$ is the Boltzmann constant ($m^2\,kg\,s^{-2}\,K^{-1}$), $T$ is the temperature (°C), and $\mu_{\text{sw}}$ is the dynamic water viscosity ($kg\,m^{-1}\,s^{-1}$) (Supplementary Table 6).

The transformation rate between different types of plastics $\tau_{\text{trans}}$ ($s^{-1}$) is calculated as:

$$\tau_{\text{trans}} = \delta\frac{\frac{dV_{\text{bf}}}{dt}}{\Delta V} \qquad (22)$$

where $\delta$ is an adjustable coefficient tuned to match the result of Kooi et al.[33], and $\Delta V$ is the deviation between the volumes of two plastic types, e.g.:

$$\Delta V(\text{PE}_{\text{neutral}}, \text{PE}_{\text{floating}}) = V_{\text{PE}_{\text{neutral}}} - V_{\text{PE}_{\text{floating}}} \qquad (23)$$

## Model sensitivity

We test the sensitivity of model results to key model parameters, including the rates for fragmentation, abrasion, beaching, biofouling, and sedimentation processes, and the magnitude of direct ocean sources (including fishing and shipping activities). We also consider additional members with different assumptions regarding the beaching and fragmentation processes (Table 1). High- or low-end value is considered for each model parameter based on value ranges reported in the literature. All the model scenarios are driven by the same emission scenario (the Middle one).

## Model ensemble

An ensemble of the model ($n = 50$) is constructed by a Monte Carlo approach by randomly generated model parameters based on their ranges reported in the literature. The fragmentation, biofouling, and sedimentation rates, which are reported to vary a wide range (i.e., no clear upper bound with a lower bound of zero), are assumed to follow a log-normal distribution, while the beaching rate and the fraction of ocean sources, which are reported to have an upper bound, follow a normal distribution. The parameter values of the test case simulation are chosen as the means, and the standard deviations are chosen as a quarter of the literature-reported ranges, i.e., the reported ranges represent 95% confidence intervals (Supplementary Fig. 3 and Supplementary Table 7). We also include another two models (#51 and #52) that sample the lowest/highest parameters. The parameters of #51 (e.g., the lowest marine source and the highest beaching rate) lead to the lowest surface plastic mass while the parameters of #52 (at the opposite end of #51 for all parameters in their ranges) lead to the highest surface plastic mass. All the ensemble members are also driven by the Middle emission scenario.

A super ensemble of the model is built by driving the above-mentioned 52 members by all the three emission inventories (i.e., High, Middle, and Low), resulting in a total number of members of

$3 \times 52 = 156$. All the sensitivity and (super) ensemble member models are run for 69 years, and the modeled surface ocean plastic mass in the last year is chosen as a metric to compare different model scenarios and members.

## Optimal estimation

A super ensemble three-dimensional variational method is used to optimally constrain the prior estimate of the global marine plastics emission by the observed surface ocean plastic masses and derive the associated emission uncertainty[81]. To account for the large uncertainties of the global marine plastic emission and modeled transport and transformation processes, we use the super ensemble ($N = 156$) as described above. For the $i$th ensemble member, the optimal estimation of the emission $E_i^{\text{a}}$ can be obtained by minimizing the cost function $J$:

$$J(E_i) = \frac{(E_i - E_i^{\text{f}})^2}{\sigma_{\text{E}}^2} + \sum_{y=1}^{Y}\frac{(y_y^{\text{o}} + \varepsilon_y^{\text{o}} - h(E_i))^2}{\sigma_{\text{o},y}^2} \qquad (24)$$

in which the superscript f denotes the prior, a denotes posterior, subscript $y$ denotes the $y$th observation with $Y$ counts in total, $y^{\text{o}}$ is the observed surface ocean plastic mass concentrations, $h$ is the observation forward operator that transforms the state variable to the observed quantities, $\sigma_{\text{E}}^2$ and $\sigma_{\text{o}}^2$ are error variances of the prior emission and observations, respectively. The observation error variance is the variance of observed quantities within a model grid for each observation, which covers the uncertainty of previous estimations of the overall floating plastic mass based on the observed data[11–13]. To represent the observation uncertainty and avoid the collapse of ensemble spread due to the same observations used to constrain the estimated emission, observations with perturbations $\varepsilon^{\text{o}}$ that is a random draw from $N(0, \sigma_{\text{o}}^2)$, are used for each ensemble member[82]. The cost function $J$ is a combination of the distance to the prior emission estimate and all available observations. By minimizing the cost function $J$ for the $N$ members separately, an ensemble of optimally estimated global marine plastic emission $\{E_i^{\text{a}}, i = 1, \ldots, N\}$ is achieved.

## Data availability

All data are available in the main text, the supplementary materials, or the web site: Plastic discharge data: https://www.ebmg.online/plastics/plastic-discharge. Observed plastic database: https://microplastics.springeropen.com/A-multilevel-dataset-of-microplastic-abundance/. Stokes drift velocity: https://tds0.ifremer.fr/thredds/GLOBCURRENT/Stokes.

## Code availability

All model code is available at the research group website: https://www.ebmg.online/plastics/MITgcm-code.

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

## Acknowledgements

We thank Rong Ji and Wei Chen for the helpful discussions and suggestions. Y.Z. is financially supported by the National Key R&D Program of China grant 2019YFA0606803, the National Natural Science Foundation of China grant 42177349, the Frontiers Science Center for Critical Earth Material Cycling, the Fundamental Research Funds for the Central Universities grant 14380168 and 14380188, and the Collaborative Innovation Center of Climate Change, Jiangsu Province.

## Author contributions

Conceptualization: Y.Z., L.L., E.Y.Z.; Methodology: P.W., R.X., Xuantong W., Y.P., Q.P., Xinle W., L.M., R.W., H.L., Xiaotong W., A.L., E.C., X.X., H.S., S.Z.; Investigation: P.W., R.X., Xuantong W., L.L. Visualization: P.W., R.X., Xuantong W.; Funding acquisition: Y.Z.; Project administration: Y.Z. Supervision: Y.Z.; Writing—original draft: Y.Z., P.W., R.X., Xuantong W., L.L.; Writing—review & editing: Y.Z., L.L., A.T.S., R.W., H.L., A.L., E.C., X.X., H.S., E.Y.Z.

## Competing interests

The authors declare no competing interests.
