## [Peer Review File · Nature Communications]

Plastic waste discharge to the global ocean constrained by seawater observationsReviewer #1 (Remarks to the Author):

This paper proposes a method to determine more accurately the amount of plastic discharge to the ocean, based on ocean transports models run for a large ensemble of model parameters controlling the different processes affecting plastic particles (vertical velocity, drifting, fragmentation, beaching, biofouling).

One discharge scenario among the 3 tested provides much better agreement with the available measurements of plastic concentrations at the ocean surface, reducing the current uncertainty about the annual plastic discharge by more than one order of magnitude.

The aim of the paper is obviously very interesting.

The method is clearly new in this field, original and appropriate. It is clearly described, and some limitations are discussed. Its implementation represents an enormous amount of work.

However I have 2 major comments that should be addressed, and could greatly increase confidence in the results.

Major comments

I have a major issue with the validation of the model only through surface observations. The processes controlling the vertical transport of plastics are largely unknown and thus particularly challenging to parameterize, even within the large error bars allowed in the parameters tested herein.

I am not really convinced that the way the best input scenario is found with this method cannot be a consequence of a wrong parameterization of the vertical transport of plastics in the model.

There could be some compensation between wrong input scenario and wrong vertical export that would lead to a best agreement with surface observations, but totally wrong plastic concentrations in the water column.

So I really think the few measurements of plastic concentrations in the water column should have been used to validate the results a posteriori, or in the optimization method if possible.

Several observations have not been published that could be used :

- Egger, M., Sulu-Gambari, F., Lebreton, L., 2020: First evidence of plastic fallout from the North Pacific Garbage Patch. *Sci Rep* 10, 7495 (2020).

<https://doi.org/10.1038/s41598-020-64465-8> .

- Pabortsava, K., Lampitt, R.S., 2020: High concentrations of plastic hidden beneath the surface of the Atlantic Ocean. *Nat Commun* 11, 4073 (2020).

<https://doi.org/10.1038/s41467-020-17932-9> .

I do believe the method is very worth being used to tackle this issue, but I am afraid the surface validation is not sufficient, and I strongly suggest to try to validate the results in the water column.

The second major concern is about the use of currents from a very coarse ocean model (2°) to transport the plastics, and the neglected influence of Stokes drift. There are several papers that have shown the large influence of ocean eddies and Stokes drift in the connection between the large sources of plastic from Asian countries towards the convergence zone of the South Indian basin, and alternatively to the South Pacific or the South Atlantic basins

- Maes, C., N. Grima, B. Blanke, E. Martinez, T. Paviet-Salomon, T. Huck, 2018: A surface "super-convergence" pathway connecting the South Indian Ocean to the subtropical South Pacific gyre. *Geophys. Res. Lett.*, 45, (4) 1915-1922, 2017GL076366, doi: 10.1002/2017GL076366 .

- van der Mheen, M., C. Pattiaratchi, E. van Sebille, 2019: Role of Indian Ocean Dynamics on Accumulation of Buoyant Debris. *JGR Oceans*, 124, 2571-2590, doi: 10.1029/2018JC014806 .

- Dobler, D., T. Huck, C. Maes, N. Grima, B. Blanke, E. Martinez, F. Arduin, 2019: Large impact of Stokes drift on the fate of surface floating debris in the South Indian Basin.

Mar. Poll. Bul., 148, 148, 202-209, doi: 10.1016/j.marpolbul.2019.07.057 .

If the model dynamics is not sufficiently good, the uncertainty on the transport processes could degrade the ability of the method to identify the best discharge scenario. This should be discussed somewhere.

Minor comments

Lines 30, 140: Mt needs to be defined very early in the article and in the abstract: I believe it is metric tons but it could be interpreted as million tons or Mega tons.

Lines 74-78: A few recent papers investigating the 3D distribution of various plastics have not been cited:

- Huck, T., R. Bajon, N. Grima, E. Portela, J.-M. Molines, T. Penduff, 2022: Three-dimensional dispersion of neutral "plastic" particles in a global ocean model. Front. Anal. Sci., 2:868515, doi: 10.3389/frans.2022.868515 .

- Richon, C., T. Gorgues, I. Paul-Pont, C. Maes, 2022: Zooplankton exposure to microplastics at the global scale: Influence of vertical distribution and seasonality. Front. Mar. Sci., 9:947309, doi:10.3389/fmars.2022.947309 .

Line 115: ref to Wilcox et al. 2020 is probable Fig. 1, not S1.

Fig. 2: Maybe you should say precisely how is r^2 computed, on the actual concentrations or on their \log_{10} .

Line 264: what does NJU-MP model means?

line 300: I don't know what is 'experiential equations'? At this point, I was expecting the equations for plastic tracers concentrations, but I wonder why the 3D 1D and 2D processes appear to be separated. What is particular to this model formulation compared to Mountford and Morales Maqueda 2019 or Richon et al 2022 (full reference above) models?

line 317: HDI could be explained.

paragraph from lines 322 to 333: I found it difficult to follow all the details of the scaling at this point.

line 338: I wonder if the China Plastic 2014 reference is available in english... if it is, please provide a link.

lines 399-401 : I cited above a reference from Dobler et al. 2019 in Marine Pollution Bulletin that shows a very striking large scale influence of Stokes drift, not really in agreement with the statement that 'the contribution ... is rather limited'.

line 510: matric > metric?

Reviewer #2 (Remarks to the Author):

Review of "Plastic waste discharge to the global ocean constrained by seawater observations" by Zhang et al.

The present study proposes a top-down approach to estimate the uncertainty of the global plastic discharge into the ocean to only one order of magnitude, evaluated to 0.72 with an uncertainty between 0.25 to 2.14 million Mt per year, whereas previous studies reported between two or three orders of magnitude. The significance of this evaluation

is important for the response in terms of the mitigation strategies that could be developed. The strategy to obtain this evaluation is based on seawater plastic observations and an ensemble of transport estimates, that include a 3D Euler-based global ocean transport and a comprehensive representation of several processes implied in the plastic cycle of life in the ocean. The approach is original and it puts attention to the complexity and the large number of parameters that are relevant to the plastic waste discharge at the global scales. The results are very interesting and they deserve publication but there are some places for improvement.

Among major comments I will recommend to consider these two types.

The presentation of the model is very difficult to follow and a lot of choice, variable and parameters is given "ad hoc" or extracted from other studies (or closed, such as equation 16 that is not strictly equivalent to the equation presented by Kooi et al.; why?). Does the variable u detailed in the "Sinking and rising" and "drifting" parts is the same field? In the former part, the balance obtained on eq 4 is based on 3 three forces but one term disappeared in the drifting part. A much more presentation of what is new and what it is based on previous studies (with adequate referring) would represent a real improvement, and will facilitate the reading and understanding. The model is complex but it does not explain why 54 different plastic tracers are considered (does the comparison with observations is polymer-dependent?); why the Stokes drift is not included when Dobler et al. (2019, MPB) found that the subtropical convergence region in the Indian Ocean moves from the east to the west of the basin, so particles leak to the South Atlantic rather than the South Pacific (even if observations are rare in the Indian basin it deserves a larger attention); why the choice of naming the different classes of biological attachment "clean, balanced and sinking" is proposed where "floating, neutral and sinking" distinction is usually presented in the literature (or explain the distinctions if any in such a case), and finally what are the arguments for the specification of the distribution type for the different processes (fragmentation, biofouling... in page 15). A simplified presentation and organization will be welcome at this stage. A last point concerns the 3D model results that are not discussed at all; even if vertical observations are really sparse at the global scales, there is the study by Pabortsava and Lampitt in 2020 (Nature comm.) that allow to make such a comparison in the Atlantic. The way the model reproduces the vertical structures may have important consequences on concentration at the surface, this should be definitely discussed.

The choice is made to deliberately limit the data set to the debris identified visually. How this choice influences the sensitivity of the results and the different estimates is not really discussed. Moreover, the spatial and temporal repartition of such (partial) observed surface plastic concentration may have some importance on the estimates due to its large heterogeneity at the global scale, but this is not presented or at least discussed neither. It is probably easier to compare a smoother field of plastic concentration with the mean state obtained by the present model. The horizontal resolution of the model is also another crucial limitation (at the exception of the computational costs) that is not really discussed in the point of view of the observations; this latter field is based on cruises and different legs that are necessarily impacted by the eddy mesoscale and sub-mesoscale variability of the ocean dynamics, processes that are not resolved by a $2^\circ \times 2.5^\circ$ model. How the data reported by Brach et al. (2018 MPB) near anticyclonic and cyclonic eddies will compared to the present coarse model, and what is the error in terms of sampling? Once again, the importance of the eddy variability, and its potential to increase the heterogeneity of the plastic concentration at sea, has been largely underlined in recent years and most of the studies, cited by this study too, are based on models with resolution (much) better than 1° . If these models are able to concentrate more plastic in the core of the convergence zones it may change the dispersion diagrams and the final results. Discussing the importance of the model-dependency is clearly one issue that deserve more attention.

To conclude, the approach is interesting and original, but the general description of the

methodology is not clear and synthesized enough to underline the important processes and to reinforce the major result on the OMs. The discussion should be improved to respond to the above points. At the present time, I recommend to send back to the authors their manuscript for major revisions.

Minor points.

- Abstract: please define "the biogeochemical rates of plastics"
- Page 1 line 40: quantify the term "smaller"
- Page 2 2nd paragraph: what's about the reservoir role of rivers as documented by van Emmerik et al. (Frontiers, 2022)?
- Page 2 line 25: I do not understand where comes the 2000 number; Cozar et al. based their study on 141 sites... please explain
- Page 3 top: there are much more study on numerical models, please cite a review or only the original ones, and quantify "substantial uncertainties"??
- Please mention the first date of observations to compare with the model simulation
- Does the surface data are based on Manta nets or else?
- Page 3 line 22: define here the "super ensemble"
- Page 3 line 31: total plastics mean MPs and macro plastics?
- Page 3 line 37: please avoid to cite some report or summary of conference that are grey literature, or give at least the internet links to access them.
- Page 4 legend of figure 2: please mention the nature of the model field (mean over the full simulation, annual mean of the last year, or else...)
- Page 5 end of the 2nd paragraph: what's about the results from other models, such as the ones cited on page 3.
- Page 5 line 21: specify the nature and the order of magnitude of "uncertainties"
- Page 5 lines 32-37: these lines are not clear and define the "uncatchable plastic" – consider to rewrite and specify the importance of this part.
- Page 6 top: it is nice to see the results for the biofouling; consider to add a table to summarize the same results for the different scenario/process. How the present results compare to the Kvale et al. study?
- Page 6 table 1: where are used and discussed the "marine sources";
- Page 7 top: How did you explain the difference between the beaching (that vary by 4 for the rate) and the sedimentation rate that vary by 2 OMs?
- Page 7 line 16: the term "potentially" suggests that it may not be the case? Comment on that please.
- Page 8 figure 4: consider to improve figure 4; I do not understand what is the y-axis unit? It is not evident to distinguish the 150 orange lines, and the thicker black is corresponding to the 0.72 value, correct? What is the reference to assume that the distribution should be a log-normal one?
- Page 8 line 22: there is no circles on figure S4a, and I do not see any differences with the 2b and 2e parts of figure 2; please clarify and add some comments to help the reader.
- Page 9 line 23: why 54 plastic tracers are considered, and more importantly, what are the impacts of such large number of classes. This point is not discussed at all.
- Page 9 "general description": what's about the settling velocity of the different polymers considered?
- Page 10: please define acronyms such as HDI or GDP
- Page 10 line 24: does the data proposed by Hoornweg and Bhata-Tata are available?
- Page 10 line 38: does the numbers of consumption of different plastic types given for 2013 apply for the whole period?
- Page 11 Eq. 6: it seems that the Cd dependency to Re is based on a specific context; please give the reference or explain how these numbers and relationships have been obtained.
- Page 11 line 22: give more details on the "few techniques"
- Page 11 line 24: "in the seawater column" the term u as a sinking or rising velocity is not usual, I will prefer to see a w. How the different types of polymer is considered in this computation should be more specified and detailed.

- Page 12 lines 20-22: what's about the surface level? Most of the diffusion process involved a laplacian or a bi-laplacian operator with some constant coefficients. Consider to give the magnitude of these coefficients and to clarify the importance of the "random walk" in such a coarse model.
- Page 12 line 35: specify the year for the reference to Lebreton et al.
- Page 13 eq 14: Do not understand how this relationship is found; in line 13 the mention of the reference Biber et al. is unclear because these latter authors based their study of different types of polymers, and not to real plastics found in polar or tropical regions; please clarify.
- Page 13 line 33: explain why there is a so large variations in the percentage of sandy shorelines
- Page 13 line 39-40: the reference to the atmospheric chemistry is not really relevant here.
- Page 15 table S6: give the sources or reference of parameters such as mortality rate, algae density or shear rate
- Page 15 line 25: metric?
- Page 15 eq 24 and line 35: what is the term "a" exactly?
- SI Fig S1: could we detect an abrupt decrease associated with MARPOL in the computation based on Weiss scenario?
- SI Fig S4: please clarify as compared to figure 2 in the main text.
- I am not sure if the table S8 is referenced in the main text; please verify. Also, give the status of fibers, if they are considered or not. Does the 194 number for Cozar et al. should be 1943?

Reviewer #3 (Remarks to the Author):

Review: Plastic waste discharge to the global ocean constrained by seawater observations

This paper attempts to constrain the highly uncertain estimates of plastic emissions into the ocean using observations of floating plastic debris combined with an ocean transport model.

The magnitude of plastic emissions into the ocean is an important research question that the community is trying to solve for years. Given the uncertainty in parameterization of the plastic emissions and the lack of direct observations, the author's approach to use observations of floating plastic to constrain the input is very promising. Reducing the uncertainty in the input estimates to 0.25–2.14 mio Mt/yr would be a significant step forward. I have read this paper with great interest.

I have two main concerns with this work:

1) The model described in this work is constrained only using surface ocean plastic observations. In order fit the surface observations, the model will transfer the entire excess plastic towards the remaining reservoirs. There needs to be a comparison with existing observations, or at least a discussion how realistic the amount of plastic in these additional reservoirs is.

As an example, models of plastic beaching have shown that the modeled amount of beached plastic is orders of magnitude higher than what is observed (see for instance Onink et al. 2021 (<https://iopscience.iop.org/article/10.1088/1748-9326/abecbd/meta>), Ryan et al. 2020 (http://www.scielo.org.za/scielo.php?pid=S0038-23532020000300003&script=sci_arttext&tlng=es) and Collins et al. 2019 (https://www.sciencedirect.com/science/article/pii/S0025326X18308798?casa_token=Otjls0MuVIsAAAAA:dMTajoiYmt2D6RFBypwD4rhgu3Mvcog5d7TNsd74OJHFIXUxartVBgcSfXdITNbcP_4IOISdRg)) when using the Lebreton or Jambeck input estimate. This is even the case when considering only 1 year of plastic emissions and using a lower

beaching rate than in the Lebreton 2019 box model. I therefore think that the present study likely also overestimates the amount of beached plastic, and potentially also the amount of plastic in other reservoirs.

2) I think that the range of at least two crucial model parameters, the fragmentation rate and the beaching rate, has been chosen too narrow and as a result the model may underestimate the full uncertainty of the input emissions. Please find more details in the text below.

Additionally, the model does not include interaction with zooplankton or aggregates, which may increase the transport out of the surface ocean. There should be a discussion how these missing processes may affect the uncertainty envelope of the input emissions.

I hope this helps! Congratulations on the paper.
Charlotte Laufkötter

Line by line comments:

Abstract line 30: Please add that the 0.72 mio Mt/yr estimate is for present day (I think?)

Abstract Line 31ff: "We find the uncertainty of the estimated global ocean plastic emissions is mainly caused by that of model parameters, indicating that a more accurate measurement of the biogeochemical rates of plastics in the ocean is required."

I think the authors mean "biogeochemical rates of plastic losses from the surface ocean" ?

Fig.1: "Lebreton and Andrady (2019) include both riverine and coastal zone sources" - I may be mistaken, but I think the paper follows the Jambeck approach which would be only coastal zone sources. Please double check.

L66: "covering all the major ocean basins" - True, however the majority of observations is from the North Atlantic and Pacific oceans. There seems to be only one cruise in the entire South Pacific and Indian Ocean.

L83: Which "important processes" does the model include? Would be good to mention here

L87: grammar
L 91/92 grammar

L93: I am assuming that a substantial uncertainty is added when introducing from numbers to mass. See for instance Roebroek et al. 2022 (<https://www.sciencedirect.com/science/article/pii/S0269749122011629>) where this problem is discussed for riverine plastic observations. Maybe this should be discussed? I don't know what typical values for the number concentrations are in your conversions, the uncertainty is reduced of course with large sample sizes.

L 112: The median Jambeck estimate is 8 mio tons, yet the high scenario is only 7.1 mio tons. Was only the lower end of the Jambeck estimates used? If so, why?

Fig. 2: The match between model and observations does not seem particularly strong, with often a mismatch of 2 orders of magnitude (same for Fig. S4). The R2 value is not very high, there are no other metrics given. I am assuming that I am looking at a point cloud consisting of 5000 measurements with most of the dots overlapping. It's hard to

judge how good the model/data fit is. Please add at least the bias.

Also Fig. 2 and S4: I am not sure why the color doing showing the latitude is of interest here - there does not seem to be a pattern with certain latitudes matching better than others?

I also wonder: The overall mass of floating plastic debris is estimated to be 14-236 thousand tons. How was this uncertainty included in the model design / optimization?

L 127-133: Please add an observational estimate for the amount of floating plastic in the respective regions.

L 156: Please clarify that this sentence does not describe the model but the current state of knowledge.

L 156ff: I appreciate these sensitivity tests! I do believe though that the fragmentation rates have been chosen too high (or maybe that a scenario with lower fragmentation rates should be added). See comments below.

L 186: The Lebreton 2019 and the Chenillat papers are cited as estimates for the amount of plastic at beaches, almost like a model evaluation.

However, I believe the Lebreton paper does not actually show that: The Lebreton study assumes a high plastic input and optimizes a simple box model such that the model matches the amount of floating surface plastic, similar to what has been done in this study. In order to match the surface plastic concentration, the model has no other option than to transport the remaining plastic to beaches or sediments, but without checking if that is a realistic outcome. The Chenillat paper likewise does not evaluate their modeled beaching results. In addition, the beaching parameterization they have used is unrealistically high - see discussion in Onink et al. 2021. Based on the results in Onink et al. 2021 and also Ryan et al. 2020 and Collins et al. 2019 I believe that these models severely overestimate beaching of plastic. See also comments below.

L 201: Given the large uncertainty associated with almost all model parameters and also the fact that some processes (such as interaction with zooplankton or marine aggregates) are missing in the model, I am not convinced that it is so unlikely that all parameters are at the higher or lower end. It would at least be important to know how large the collective variability is. Please add scenarios with all parameters assuming highest/lowest values.

L 408: Both the Song paper and the Efimova paper describe fragmentation of plastic on beaches, which is not directly transferable to open ocean conditions. The open ocean fragmentation is likely lower.

L413: The Lebreton model optimizes the observed surface plastic concentration when using a given plastic emission estimate, similarly to what is done in this work. I don't think this study can be used to constrain the fragmentation rate, what they report is the most likely fragmentation rate assuming a certain plastic emission. If the plastic emission they are using is off or likewise any of the other fluxes they assume such as the beaching, their numbers would change.

L 422: So what values are assumed for R? I am assuming that q is in W/m² and varies between 0-800 W/m² roughly, that would lead to R being between 10-55 %/yr. This seems like an unreasonably high fragmentation rate for the open ocean. See also Onink et al. 2022 (just published):

<https://pubs.acs.org/doi/10.1021/acs.est.2c03363>

L 428: "A plastic particle is beached when it arrives at a beach-adjacent cell". - Do you mean they always beach once they are in a beach-adjacent cell, or that there is a probability that they beach according to the beaching rate described later?

L 431: The Lebreton paper is not a good citation here: They assume that beached plastic resuspends into the ocean, but they don't actually show it. Better to cite one of the Hinata papers here.

L 432: Please mention that the Atwood paper is about microplastic (I think)

L 433 It is unclear to me how you transformed the drifter data into a beaching rate. In what distance from the shore were the drifters floating, and how long did it take until they beached?

L 436: If you assume a net beaching rate (i.e. beaching - resuspension) as you describe, you simulate beaches as a final sink for plastic. You need to double-check if the simulated overall amount of plastic on beaches is realistic. How many kg of plastic / m² does your model suggest? See for instance Onink et al. 2021 for a comparison of simulated and observed beached plastic.

L 440: Why is 10%/day a conservative beaching rate? It seems to be one possible beaching rate among many. Also, I understand that the model resolution is roughly 2 degrees, so a plastic particle entering a coastal grid cell may have to travel 200km to reach the coast, or 100km from the center of the cell. Assuming that 10% of all particles manage to reach the coast per day seems like a particularly high beaching rate to me.

Supplement Line 66: The estimates of coastal dumping only cover macroplastic (or potentially secondary microplastic), not tire wear particles or lost resin pellets. The Jambeck paper considers mismanaged municipal solid plastic waste, which does not include microplastic.

Fig S1: Please explain what you mean with "the abrupt decrease in the 1980s is associated with the MARPOL convention"

Update: I found the explanation later in the text, maybe just refer to it here

Note: *italic fonts are for comments; blue fonts are our responses with purple for quoted text from the revised manuscript.*

Reviewer #1 (Remarks to the Author):

This paper proposes a method to determine more accurately the amount of plastic discharge to the ocean, based on ocean transports models run for a large ensemble of model parameters controlling the different processes affecting plastic particles (vertical velocity, drifting, fragmentation, beaching, biofouling). One discharge scenario among the 3 tested provides much better agreement with the available measurements of plastic concentrations at the ocean surface, reducing the current uncertainty about the annual plastic discharge by more than one order of magnitude.

The aim of the paper is obviously very interesting. The method is clearly new in this field, original and appropriate. It is clearly described, and some limitations are discussed. Its implementation represents an enormous amount of work.

We thank the reviewer for the recognition of our work and the comments and suggestions for improving this manuscript.

However I have 2 major comments that should be addressed, and could greatly increase confidence in the results.

Major comments

(1) I have a major issue with the validation of the model only through surface observations. The processes controlling the vertical transport of plastics are largely unknown and thus particularly challenging to parameterize, even within the large error bars allowed in the parameters tested herein.

I am not really convinced that the way the best input scenario is found with this method cannot be a consequence of a wrong parameterization of the vertical transport of plastics in the model. There could be some compensation between wrong input scenario and wrong vertical export that would lead to a best agreement with surface observations, but totally wrong plastic concentrations in the water column.

So I really think the few measurements of plastic concentrations in the water column should have been used to validate the results a posteriori, or in the optimization method if possible. Several observations have now been published that could be used :

- Egger, M., Sulu-Gambari, F., Lebreton, L., 2020: First evidence of plastic fallout from the North Pacific Garbage Patch. Sci Rep 10, 7495 (2020). <https://doi.org/10.1038/s41598-020-64465-8>.

- Pabortsava, K., Lampitt, R.S., 2020: High concentrations of plastic hidden beneath the surface of the Atlantic Ocean. Nat Commun 11, 4073 (2020). <https://doi.org/10.1038/s41467-020-17932-9>.

I do believe the method is very worth being used to tackle this issue, but I am afraid the surface validation is not sufficient, and I strongly suggest to try to validate the results in the water column.

We thank the reviewer for bringing up this point. We added a comparison between the simulated vertical profile and the observations (Egger et al. 2020; Pabortsava and Lampitt

2020) in the supplementary information (Figure S5). Generally, our model captures the vertical trend of plastic concentrations both in the North Pacific Ocean and the Atlantic Ocean: high concentrations in the surface water and decreased concentrations with the depth. So, we believe that our model can predict the fraction of plastics in each compartment (e.g., surface, water volume, and beaches). In the Pacific Ocean (Fig. S5a), we did not compare the profiles in the deep ocean because the concentrations under the depth of 500 m are below the detection limit (Egger et al. 2020). In the Atlantic Ocean (Fig. S5b), the observed concentrations are elevated in the subsurface (e.g., 100 – 250 m) in some sites while the simulated concentrations decrease slowly in all sites. Indeed, the vertical distribution of microplastics in the water column is an emerging research field that merits further evaluation and we are considering a more detailed parameterization of this process in our recent studies. The following discussions were added in the main text:

Line 289-292: “Finally, the observations of plastic abundance in other compartments, e.g., the water column, are rather limited to evaluate the model, but our model captures the observed vertical trend, indicating a reasonable representation of the fraction of plastic mass in the surface ocean (Fig S5).”

Fig. S5. Comparison between observed and modeled vertical profile of plastic mass for the a) Pacific and b) Atlantic Ocean. The dash lines indicate individual observed profiles with the filled-colored circles representing sampling depths. The light blue solid lines are the mean plastic mass concentrations over the sampling sites by the 52 member models under the middle emission scenario (other emission scenarios simulate higher or lower concentrations but with similar vertical trends), while the dark blue solid lines are the ensemble means. The observations are from Egger et al. (2020) and Pabortsave and Lampitt (2020) for the Pacific and Atlantic Ocean, respectively.

(2) *The second major concern is about the use of currents from a very coarse ocean model (2°) to transport the plastics, and the neglected influence of Stokes drift. There are several papers that have shown the large influence of ocean eddies and Stokes drift in the connection between the large sources of plastic from Asian countries towards the convergence zone of the South Indian basin, and alternatively to the South Pacific or the South Atlantic basins - Maes, C., N. Grima, B. Blanke, E. Martinez, T. Paviet-Salomon, T. Huck, 2018: A surface “super-convergence” pathway connecting the South Indian Ocean to the subtropical South*

Pacific gyre. Geophys. Res. Lett., 45, (4) 1915-1922, 2017GL076366, doi: 10.1002/2017GL076366.

- van der Mheen, M., C. Pattiaratchi, E. van Sebille, 2019: *Role of Indian Ocean Dynamics on Accumulation of Buoyant Debris. JGR Oceans*, 124, 2571-2590, doi: 10.1029/2018JC014806.

- Dobler, D., T. Huck, C. Maes, N. Grima, B. Blanke, E. Martinez, F. Ardhuin, 2019: *Large impact of Stokes drift on the fate of surface floating debris in the South Indian Basin. Mar. Poll. Bul.*, 148, 148, 202-209, doi: 10.1016/j.marpolbul.2019.07.057.

If the model dynamics is not sufficiently good, the uncertainty on the transport processes could degrade the ability of the method to identify the best discharge scenario. This should be discussed somewhere.

We appreciate the reviewer's valuable suggestion and we added the Stokes drift in our model. We took the estimated Stokes drift velocity of other models and applied to all the plastic tracers in our model. The following sentence was added to the Method section:

Line 449-455: "Stokes drift, a near-surface velocity induced by ocean wind-generated gravity waves, can contribute to the near-surface transport of mass including plastics (Stokes, 1847). For example, it changes the locations of convergence zones for plastics mainly caused by Ekman currents, resulting in a westward entrainment in the north of the convergence zone in the Indian Ocean (Dobler et al. 2019). We consider the Stokes drift for plastic tracers on the surface ocean in the model. The surface Stokes drift estimates are from the WaveWatch III and the annual mean from 1990 to 2015 is applied for the whole simulation period (<http://globcurrent.ifremer.fr/>)."

We also included a new figure (Fig. S6) to compare the results with and without Stokes drift. The following text was also added to the supplementary text as a new subsection:

Line 80-88: "Stokes drift We take the estimated Stokes drift velocity from the WaveWatch III and apply it to all the plastic tracers in our model (<http://globcurrent.ifremer.fr/>). Only the data from 1990 to 2015 are available so we apply the climatological annual mean of this period for the whole simulation period. The strongest Stokes drift is simulated in the Southern Ocean and the high-latitude ocean in the northern hemisphere where plastic concentrations are relatively low (Fig. S6), consistent with previous results (Carrasco et al. 2014). The modeled plastics accumulation in the subtropical gyres is moved westward slightly by Stokes drift, while the plastics in the Southern Ocean are moved eastward."

(a) No Stokes

(b) With Stokes

10^{-1.0} 10^{-0.5} 10⁰ 10^{0.5} 10^{1.0} 10^{1.5} 10^{2.0} 10^{2.5} 10^{3.0} 10^{3.5} 10^{4.0} 10^{4.5} 10^{5.0}

Fig. S6. Modeled spatial pattern of surface plastic abundance: a) with Stokes drift, and b) without Stokes drift.

Minor comments

(3) Lines 30, 140: Mt needs to be defined very early in the article and in the abstract: I believe it is metric tons but it could be interpreted as million tons or Mega tons.

The reviewer's point is well taken. In the abstract, there is no need to define metric tons as Mt because the acronym does not appear in the remainder of the abstract. We did define metric tons as Mt when it first appears in the main text:

Line 70-71: "which ranges 14.4 – 236 thousand metric tons (Mt)"

(4) Lines 74-78: A few recent papers investigating the 3D distribution of various plastics have not been cited:

- Huck, T., R. Bajon, N. Grima, E. Portela, J.-M. Molines, T. Penduff, 2022: Three-dimensional dispersion of neutral "plastic" particles in a global ocean model. *Front. Anal. Sci.*, 2:868515, doi: 10.3389/frans.2022.868515.

- Richon, C., T. Gorgues, I. Paul-Pont, C. Maes, 2022: Zooplankton exposure to microplastics at the global scale: Influence of vertical distribution and seasonality. *Front. Mar. Sci.*, 9:947309, doi:10.3389/fmars.2022.947309.

These papers are cited in the revised manuscript to strengthen our discussions:

Line 76-79: “Numerical models with a variety of complexity have been developed in the past decade (Wichmann et al. 2019; Mountford and Morales Maqueda, 2019; Lebreton et al. 2019; Onink et al. 2021; Lobelle et al. 2021; **Huck et al. 2022; Richon et al. 2022**).”

(5) Line 115: *ref to Wilcox et al. 2020 is probable Fig. 1, not S1.*

Sorry for the confusion. Wilcox et al. 2020 show that the abundance of plastics at the sea surface is strongly related to cumulative global plastic production instead of the production within the same year. So, we used the trend of accumulated global plastic production as a proxy for the trend of plastic waste emission. The sentence was modified as:

Line 119-121: “According to Wilcox et al. (2020), the historical trends of emissions from 1950 to 2020 are assumed to follow the trend of accumulated global plastic production (Fig. S1).”

(6) Fig. 2: *Maybe you should say precisely how is r^2 computed, on the actual concentrations or on their \log_{10} .*

r^2 was calculated on the \log_{10} of the concentrations and we stated this precisely in the caption of Fig. 2:

“All the concentrations, and the calculation of R^2 and RMSE are in a base 10 logarithmic scale with a unit of g km^{-2} .”

(7) Line 264: *what does NJU-MP model means?*

We added the full name of the model in the main text:

Line 305-307: “We simulate the fate and transport of plastic by the NJU-MP (Nanjing University Marine-Plastic) model based on the MITgcm model framework (Marshall et al. 1997; Peng et al. 2021).”

(8) line 300: *I don't know what is 'experiential equations'? At this point, I was expecting the equations for plastic tracers concentrations, but I wonder why the 3D 1D and 2D processes appear to be separated.*

Sorry for the confusion. We modified ‘experiential equations’ as ‘empirical equations’. All the processes are occurring simultaneously in the real ocean but are simulated one by one in the model (aka operator splitting technique for numerical models). For example, in one time step, the 1D sinking/rising is computed first, followed by 2D horizontal advection and other processes. The error associated with this approximation becomes smaller if the time step is short enough. To clarify this, the sentences were modified as:

Line 341-346: “Thus, in our model, the motion of plastic particles is resolved into advection and mixing (three-dimensional), sinking/rising (one-dimensional), and drifting (two-dimensional). **The above processes are computed separately in the model (aka operator splitting) though they occur simultaneously in the real ocean.** We calculate particles' sinking, rising, and drifting velocity by fluid dynamics and **empirical equations.**”

(9) *What is particular to this model formulation compared to Mountford and Morales Maqueda 2019 or Richon et al 2022 (full reference above) models?*

These studies developed Eulerian 3D models for marine plastics and simulated the diffusion/advection and sinking/rising of plastic particles. Our study considers more complex transport and transformation processes of plastic particles in the ocean, including drifting, fragmentation/abrasion, beaching, and biofouling/defouling. Also, we consider the historical discharge of plastics from 1950 to 2018. We modified the sentences to acknowledge these previous works:

Line 76-79: “Numerical models with a variety of complexity have been developed in the past decade (Wichmann et al. 2019; **Mountford and Morales Maqueda, 2019**; Lebreton et al. 2019; Onink et al. 2021; Lobelle et al. 2021; **Huck et al. 2022**; Richon et al. 2022).”

(10) *line 317: HDI could be explained.*

HDI means Human Development Index. We explained it in the manuscript:

Line 360-361: “A key feature of the Mai inventory is a prediction model for the waste treatment rate by Human Development Index (HDI)...”

(11) *paragraph from lines 322 to 333: I found it difficult to follow all the details of the scaling at this point.*

These sentences were modified to clarify the scaling process:

Line 367-380: “The coastal plastic emission inventory follows the framework of Jambeck et al. (2015) that use MPW as proxy data to estimate plastic discharge but has a lower estimation (Chassignet et al. 2021). Similar to the Lebreton inventory, it also relies on the solid waste data from the World Bank (Hoornweg and Bhada-Tata, 2012). There is double counting for emissions from coastal areas that are also located near river mouths, but the overall contribution is negligible (Chassignet et al. 2021). This Jambeck/Chassignet inventory is directly used for coastal emissions when the Lebreton inventory is chosen for rivers. **When the other two lower riverine inventories are chosen, the above coastal inventory is also scaled down to keep the ratio between the riverine and coastal emissions constant.** In addition to the riverine and coastal plastic emission, we consider direct ocean emissions from marine activities such as shipping and fishing, which are assumed to account for 25% of terrestrial discharge for all three emission scenarios (Macfadyen et al. 2009; Kane et al. 2020). The spatial pattern of the discharge from marine activities is allocated according to the global footprint of fisheries (Kroodsma et al. 2018) and shipping tracks (Wang et al. 2021).”

(12) *line 338: I wonder if the China Plastic 2014 reference is available in english... if it is, please provide a link.*

The China Plastic 2014 is not available in English (<http://nianjian.xiaze.com/down/2016/zgslgynj-2014.html>). However, current translate engine (e.g., Google translate) has largely removed the language barrier.

(13) *lines 399-401: I cited above a reference from Dobler et al. 2019 in Marine Pollution Bulletin that shows a very striking large scale influence of Stokes drift, not really in*

agreement with the statement that 'the contribution ... is rather limited'.

The sentence was deleted in the revised manuscript, and we added the Stokes drift process in our model. Please refer to our response to the second major comment for more details.

(14) *line 510: matric > metric?*

Revised as suggested.

Reviewer #2 (Remarks to the Author):

Review of “Plastic waste discharge to the global ocean constrained by seawater observations” by Zhang et al.

The present study proposes a top-down approach to estimate the uncertainty of the global plastic discharge into the ocean to only one order of magnitude, evaluated to 0.72 with an uncertainty between 0.25 to 2.14 million Mt per year, whereas previous studies reported between two or three orders of magnitude. The significance of this evaluation is important for the response in terms of the mitigation strategies that could be developed. The strategy to obtain this evaluation is based on seawater plastic observations and an ensemble of transport estimates, that include a 3D Euler-based global ocean transport and a comprehensive representation of several processes implied in the plastic cycle of life in the ocean. The approach is original and it puts attention to the complexity and the large number of parameters that are relevant to the plastic waste discharge at the global scales. The results are very interesting and they deserve publication but there are some places for improvement.

We thank the reviewer for the recognition of our work and the comments and suggestions for improving this manuscript.

(1) Among major comments I will recommend to consider these two types.

The presentation of the model is very difficult to follow and a lot of choice, variable and parameters is given “ad hoc” or extracted from other studies (or closed, such as equation 16 that is not strictly equivalent to the equation presented by Kooi et al.; why?).

We are sorry for the confusion. We carefully checked the presentation of our model and made the description of the equations, variables, and parameters clearer. Please refer to our responses to your comments below and the revised manuscript for more details.

For equation 16, we indeed adapted their original equation due to the different model setting ups. The largest difference is that the Kooi model is based on Lagrangian perspective, i.e., following the **trajectories** of individual particles, while ours is on Euler perspective, i.e., following the temporal changes of all the particles within a **grid box**. Indeed, the Kooi model directly tracked the growth and respiration of plankton/microbe on the particles (the second and fourth terms in the right-hand side of the following equation, respectively):

$$\frac{dA}{dt} = \frac{\beta_A A_A}{\theta_{pl}} + \mu_A(T, I)A - m_A A - Q_{10}^{(T-20)/10} R_{20} A \quad (\text{Kooi et al. 2017})$$

However, such a process is hard to be replicated in an Euler model because we don't know the history/trajectories of these particles. So, we only kept the first and third term in our model:

$$\frac{dA}{dt} = \frac{\beta_a A_a}{\theta_p} - m_a A \quad (\text{Equation 16 in the main text})$$

The growth and respiration of plankton/microbe that have an impact on the density of plastic particles are simulated with the transformation among three types of tracers: floating, neutral, and sinking. This difference was clarified in the revised manuscript:

Line 497-502: “Biofouling of light plastic types (PE and PP) is modeled following Kooi

et al. (2017) that is based on Lagrangian perspective but adjusted for our Eulerian-based framework. Kooi et al. (2017) directly tracked the growth and respiration of plankton/microbe on the particles but such a process is hard to be replicated in an Euler model because we don't know the history/trajectories of these particles. This process has an impact on the density of plastic particles and is simulated with three types of tracers in our model: floating, neutral, and sinking.”

(2) Does the variable u detailed in the “Sinking and rising” and “drifting” parts is the same field?

Sorry for this confusion. They are different. u in the “Sinking and rising” is the sinking or rising velocity while u in the “Drifting” is the drifting velocity. u in the “Sinking and rising” was modified as w in the revised manuscript.

(3) In the former part, the balance obtained on eq 4 is based on 3 three forces but one term disappeared in the drifting part.

F_D in eq 4 is the vertical dragging force by seawater while in the drift we assume that particles are floating on the surface and are affected by horizontal dragging force by seawater (F_s). We modified the following sentence to clarify this:

Line 402: “where F_D is vertical dragging force of the seawater, ...”

Line 429-430: “ F_s and F_a are the horizontal dragging force by seawater and air, respectively.”

(4) A much more presentation of what is new and what it is based on previous studies (with adequate referring) would represent a real improvement, and will facilitate the reading and understanding.

We thank the reviewer for this suggestion. We used words such as “we develop” or “we solve” to show new development of our model, and “is based on” or “we follow” to specify the part based on previous studies. We also added more references for these studies. Please refer to the revised manuscript for more details.

(5) The model is complex but it does not explain why 54 different plastic tracers are considered (does the comparison with observations is polymer-dependent?);

We elaborated on how the 54 different plastic tracers are chosen:

Line 319-329: “The model includes 5 categories of different chemical compositions and the density of each category is pre-determined: PE (polyethylene, 950 kg m^{-3}), PP (polypropylene, 900 kg m^{-3}), PVC (polyvinyl chloride, 1410 kg m^{-3}), PS (polystyrene, 1050 kg m^{-3}), and ABS (acrylonitrile butadiene styrene, 1050 kg m^{-3}). Each category has 6 size bins with bounds equally distributed on a log scale: four for microplastic: $<0.0781 \text{ mm}$, $0.0781\text{--}0.3125 \text{ mm}$, $0.3125\text{--}1.25 \text{ mm}$, and $1.25\text{--}5 \text{ mm}$, and two for macroplastic: $5\text{--}50 \text{ mm}$ and $>50 \text{ mm}$. The plastic's density is increased when biofouled and we consider three states for PE and PP: floating, neutral, and sinking (no floating and neutral states for PVC, PS, and ABS as they are originally heavier than or as heavy as seawater). **There are a total of 54 plastic tracers in the model (PE and PP each has 18 tracers while the other three each has 6 tracers)**, representing the common chemical compositions, size ranges and biofouling

states of plastics in the marine environment.”

The comparison with observations is not polymer-dependent as most of the reported measured plastic concentrations are for the total plastics. We clarified this in the revised manuscript:

Line 123: “Comparison between observed and modeled **total** surface ocean plastic mass concentrations.”

(6) why the Stokes drift is not included when Dobler et al. (2019, MPB) found that the subtropical convergence region in the Indian Ocean moves from the east to the west of the basin, so particles leak to the South Atlantic rather than the South Pacific (even if observations are rare in the Indian basin it deserves a larger attention);

We thank the reviewer for bringing it up. We added the Stokes drift in our model. We took the estimated Stokes drift velocity of other models and applied to all the plastic tracers in our model. The following sentence was added to the Method section:

Line 449-455: “Stokes drift, a near-surface velocity induced by ocean wind-generated gravity waves, can contribute to the near-surface transport of mass including plastics (Stokes, 1847). For example, it changes the locations of convergence zones for plastics mainly caused by Ekman currents, resulting in a westward entrainment in the north of the convergence zone in the Indian Ocean (Dobler et al. 2019). We consider the Stokes drift for plastic tracers on the surface ocean in the model. The surface Stokes drift estimates are from the WaveWatch III and the annual mean from 1990 to 2015 is applied for the whole simulation period (<http://globcurrent.ifremer.fr/>).”

We also included a new figure (Fig. S6) to compare the results with and without Stokes drift. The following text was also added to the supplementary text as a new subsection:

Line 80-88: “Stokes drift We take the estimated Stokes drift velocity from the WaveWatch III and apply it to all the plastic tracers in our model (<http://globcurrent.ifremer.fr/>). Only the data from 1990 to 2015 are available so we apply the climatological annual mean of this period for the whole simulation period. The strongest Stokes drift is simulated in the Southern Ocean and the high-latitude ocean in the northern hemisphere where plastic concentrations are relatively low (Fig. S6), consistent with previous results (Carrasco et al. 2014). The modeled plastics accumulation in the subtropical gyres are moved westward slightly by Stokes drift, while the plastics in the Southern Ocean are moved eastward.”

(a) No Stokes

(b) With Stokes

Fig. S6. Modeled spatial pattern of surface plastic abundance: a) with Stokes drift, and b) without Stokes drift.

(7) why the choice of naming the different classes of biological attachment “clean, balanced and sinking” is proposed where “floating, neutral and sinking” distinction is usually presented in the literature (or explain the distinctions if any in such a case),

We thank the reviewer for this suggestion. “Clean, balanced and sinking” were modified as “floating, neutral and sinking”, respectively, in the revised manuscript.

(8) and finally what are the arguments for the specification of the distribution type for the different processes (fragmentation, biofouling... in page 15).

Parameters reported to have an upper bound, e.g. the fraction of ocean sources, are assumed to follow a normal distribution while the parameters reported to vary a wide range (i.e., no clear upper bound with a lower bound of zero, e.g., fragmentation rate) are assumed to follow a log-normal distribution. We clarified this in the revised manuscript:

Line 548-552: “The fragmentation, biofouling, and sedimentation rates, which are

reported to vary a wide range (i.e., no clear upper bound with a lower bound of zero), are assumed to follow a log-normal distribution, while the beaching rate and the fraction of ocean sources, which are reported to have an upper bound, follow a normal distribution.”

(9) *A simplified presentation and organization will be welcome at this stage.*

We rewrote some paragraphs in the Materials and Methods Section. Please refer to the revised main text for more details.

(10) *A last point concerns the 3D model results that are not discussed at all; even if vertical observations are really sparse at the global scales, there is the study by Pabortsava and Lampitt in 2020 (Nature comm.) that allow to make such a comparison in the Atlantic. The way the model reproduces the vertical structures may have important consequences on concentration at the surface, this should be definitely discussed.*

We thank the reviewer for bringing up this important aspect. We added a comparison between the simulated vertical profile and the observations (Egger et al. 2020; Pabortsava and Lampitt 2020) in the supplementary information (Figure S5). Generally, our model captures the vertical trend of plastic concentrations both in the North Pacific Ocean and the Atlantic Ocean: high concentrations in the surface water and decreased concentrations with the depth. So, we believe that our model can predict the fraction of plastics in each compartment (e.g., surface, water volume, and beaches). In the Pacific Ocean (Fig. S5a), we did not compare the profiles in the deep ocean because the concentrations under the depth of 500 m are below the detection limit (Egger et al. 2020). In the Atlantic Ocean (Fig. S5b), the observed concentrations are elevated in the subsurface (e.g., 100 – 250 m) in some sites while the simulated concentrations decrease slowly in all sites. Indeed, the vertical distribution of microplastics in the water column is an emerging research field that merits further evaluation and we are considering a more detailed parameterization of this process in our recent studies. The following discussions were added in the main text:

Line 289-292: “Finally, the observations of plastic abundance in other compartments, e.g., the water column, are rather limited to evaluate the model, but our model captures the observed vertical trend, indicating a reasonable representation of the fraction of plastic mass in the surface ocean (Fig S5).”

Fig. S5. Comparison between observed and modeled vertical profile of plastic mass for the a)

Pacific and b) Atlantic Ocean. The dash lines indicate individual observed profiles with the filled-colored circles representing sampling depths. The light blue solid lines are the mean plastic mass concentrations over the sampling sites by the 52 member models under the middle emission scenario (other emission scenarios simulate higher or lower concentrations but with similar vertical trends), while the dark blue solid lines are the ensemble means. The observations are from Egger et al. (2020) and Pabortsave and Lampitt (2020) for the Pacific and Atlantic Ocean, respectively.

(11) The choice is made to deliberately limit the data set to the debris identified visually. How this choice influences the sensitivity of the results and the different estimates is not really discussed.

We added the following discussions in the revised supplementary information:

Line 33-45: “Visual inspection is known to lead to an overestimation of the particle count because 20-70% of the particles identified visually as plastics might be of other chemical compositions, e.g., coal ash, especially for the particle smaller than 500 μm (Eriksen et al. 2013; Hidalgo-Ruz et al. 2012). However, visual identification misses the transparent and small microplastics thus underestimating the plastic concentrations (Song et al. 2015). The data obtained by different methodologies are thus generally not comparable with each other. More importantly, different methodologies are used in different studies covering different ocean basins. A mixture of data obtained by different methodologies may distort the real spatial pattern of surface ocean plastic abundance and reduce the reliability of the comparison between simulations and observations. There is a trade-off between “more data” and “ghost spatial pattern” arising from different measurement methodologies. Therefore, we use only the dataset obtained by Neuston net and the visual identification method due to its large sampling number and spatial coverage.”

(12) Moreover, the spatial and temporal repartition of such (partial) observed surface plastic concentration may have some importance on the estimates due to its large heterogeneity at the global scale, but this is not presented or at least discussed neither.

We totally agree with the reviewer about the large heterogeneity of the plastics at the global scale and that the observational data has a large impact on our results. Indeed, most of the observed data used in this study were obtained during 2000 – 2015 by “sporadically” cruise studies that are often not repeated. They are also mostly located in the centers of gyres. The lack of long time series data makes it hard to detect a robust temporal trend. More samplings in oceans with lower plastic concentrations such as the Southern Ocean are also needed. The following discussions were added in the revised manuscript:

Line 285-289: “Also, our results are dependent on available observed surface plastic abundance datasets during 2000 – 2015 by “sporadically” cruise studies that are often not repeated and mostly located in the centers of gyres. More observed data that cover the global ocean (e.g., over the southern hemisphere) and a longer time period can increase the reliability of our optimal estimation.”

(13) It is probably easier to compare a smoother field of plastic concentration with the mean state obtained by the present model.

We thank the reviewer for this suggestion. We used the original observed data instead of interpolating them into a smoother field to illustrate the spatiotemporal variability (or heterogeneity) of the observations.

(14) The horizontal resolution of the model is also another crucial limitation (at the exception of the computational costs) that is not really discussed in the point of view of the observations; this latter field is based on cruises and different legs that are necessarily impacted by the eddy mesoscale and sub-mesoscale variability of the ocean dynamics, processes that are not resolved by a 2°x2.5° model. How the data reported by Brach et al. (2018 MPB) near anticyclonic and cyclonic eddies will compared to the present coarse model, and what is the error in terms of sampling?

Please refer to our response to the comment #15 below.

(15) Once again, the importance of the eddy variability, and its potential to increase the heterogeneity of the plastic concentration at sea, has been largely underlined in recent years and most of the studies, cited by this study too, are based on models with resolution (much) better than 1°. If these models are able to concentrate more plastic in the core of the convergence zones it may change the dispersion diagrams and the final results. Discussing the importance of the model-dependency is clearly one issue that deserve more attention.

We totally agree with the reviewer and are fully aware of the limitations. Indeed, higher-resolution model has a better representation of the ocean dynamics but cannot effectively explore the variability caused by different biogeochemical processes. There is definitely a trade-off between them given a limited computational cost. The sensitivity experiments and the large model ensemble in this study allow us to assess the uncertainty of many processes and constrain the plastic emissions. Also, our model focuses on the global ocean so the impact of small-scale processes such as eddies seems to be of second-order importance. We added some discussions in the revised manuscript:

Line 162-174: “The model has large uncertainties due to the relatively coarse resolution and simplified ocean physics. The effects of some small-scale processes such as anticyclonic and cyclonic eddies are not included in the model. Brach et al. (2018) found that the microplastic concentrations are 9.4 times higher in an anticyclonic eddy than in a cyclonic eddy. In addition, the present understanding of the physical and biogeochemical processes of plastics in the ocean is incomplete. There is indeed a trade-off between a finer resolution and a larger number of sensitivity experiments. We set many model scenarios to test the sensitivity of model results to the model parameters (Table 1 and Fig. 3a), which allows us to assess the model uncertainty and compare it against that of the plastic emissions. These model scenarios have different fragmentation, abrasion, beaching, biofouling, and sedimentation rates with values ranging as reported in the literature but are driven by the same emission scenario (the Middle one). Also, our model focuses on the global scale so the impact of small-scale processes seems to be of second-order importance.”

(16) To conclude, the approach is interesting and original, but the general description of the methodology is not clear and synthetized enough to underline the important processes and to reinforce the major result on the OMs. The discussion should be improved to respond to the

above points. At the present time, I recommend to send back to the authors their manuscript for major revisions.

We thank the reviewer again for the recognition of our approach. We rearranged the Materials and Methods Section and added more discussions on uncertainties as suggested. Please refer to the response to the above comments and the revised manuscript for more details.

Minor points.

(17) Abstract: please define “the biogeochemical rates of plastics”

The biogeochemical rates of plastics mean the rates of transport and transformation of plastic particles in the ocean. The sentence was modified as:

Line 30-33: “We find the uncertainty of the estimated global ocean plastic emissions is mainly caused by that of model parameters, indicating that a more accurate measurement of **the rates of plastic transport and transformation** in the ocean is required.”

(18) Page 1 line 40: quantify the term “smaller”

The sentence was modified as:

Line 38-41: “Ocean plastics mainly derive from terrestrial sources via either riverine discharge or the erosion of waste from coastal zones with a smaller contribution from direct dumping by shipping and fishing activities (~**25% of the total terrestrial discharge**) (Li et al. 2016; Jambeck et al. 2015; Lebreton et al. 2017).”

(19) Page 2 2nd paragraph: what’s about the reservoir role of rivers as documented by van Emmerik et al. (Frontiers, 2022)?

The following discussions were added in the 2nd paragraph of Page 2:

Line 56-57: “Also, rivers act as reservoirs for plastics, affecting the discharge to oceans (van Emmerik et al. 2022).”

(20) Page 2 line 25: I do not understand where comes the 2000 number; Cozar et al. based their study on 141 sites... please explain

Sorry for the mistake. Here we wanted to show that there are more than 3000 samples in the dataset assembled by Cozar et al. (2014). We corrected it in the revised manuscript:

Line 69: “There are more than 3000 samples of surface net tow data, ...”

(21) Page 3 top: there are much more study on numerical models, please cite a review or only the original ones, and quantify “substantial uncertainties”??

We added two more citations: Huck et al. 2022; Richon et al. 2022. The uncertainties are 2-3 orders of magnitude (van Sebille et al. 2015). Now the sentences read as:

Line 76-81: “Numerical models with a variety of complexity have been developed in the past decade (Wichmann et al. 2019; Mountford and Morales Maqueda, 2019; Lebreton et al. 2019; Onink et al. 2021; Lobelle et al. 2021; Huck et al. 2022; Richon et al. 2022). These models track the spatial pattern relatively well but contain substantial uncertainties of 2-3 OMs in estimating the surface ocean plastic abundance (van Sebille et al. 2015).”

(22) *Please mention the first date of observations to compare with the model simulation*

The observations are from 2000 to 2015. We added the date in the revised manuscript:

Line 90-91: “The model results are constrained by measured surface ocean plastic abundance sampled between **2000 and 2015.**”

(23) *Does the surface data are based on Manta nets or else?*

The surface data are based on Neuston net (Table S8).

(24) *Page 3 line 22: define here the “super ensemble”*

The super ensemble includes the 52 member models driven by the 3 emission scenarios ($156 = 52 \times 3$). The sentence was modified as:

Line 101-103: “A super ensemble **containing 156 (= 52 × 3) members** is constructed by sampling the uncertainties of modeled transport and transformation processes ($n = 52$) and the existing emission inventories ($n = 3$).”

(25) *Page 3 line 31: total plastics mean MPs and macro plastics?*

The total plastics mean the sum of micro- and macroplastics. We clarified it in the main text:

Line 110-112: “The Lebreton and Mai inventories are for total plastics, **including both microplastics (diameter < 5 mm) and macroplastics (diameter > 5 mm)**, while the Weiss inventory only includes the emission of microplastics.”

(26) *Page 3 line 37: please avoid to cite some report or summary of conference that are grey literature, or give at least the internet links to access them.*

The reference to the summary of a conference was deleted.

(27) *Page 4 legend of figure 2: please mention the nature of the model field (mean over the full simulation, annual mean of the last year, or else...)*

The model field is the annual mean of the last year and we clarified this in the caption:

Line 125-126: “For a-c, the background colors are the modeled annual mean for the year 2018, ...”

(28) *Page 5 end of the 2nd paragraph: what’s about the results from other models, such as the ones cited on page 3.*

Most models do not consider the real emissions and are not able to compare with the observations. Figure R1 shows the comparison of the results from three models with the observations (adapted from Figure 2 from van Sebille et al., 2015). These models all underestimate the observations at low concentration ranges. They also simulate inaccurate locations for the highest plastic concentrations (to the west of observations). We added this in the revised manuscript:

Line 158-161: “Most previous models do not consider real plastic emissions thus are not comparable with observations. Some models also have issues in reproducing the observations in the low concentration ranges or simulating inaccurate ocean location for the highest plastic

concentration patches (van Sebille et al., 2015).”

Figure R1. (a) Comparison between the three ocean models and the standardized observations at each surface trawl location. The points are color coded according to basin, and the black lines are the one-to-one lines. Adapted from Figure 2 of van Sebille et al., 2015.

(29) Page 5 line 21: *specify the nature and the order of magnitude of “uncertainties”*

The paragraphs followed this line discuss the uncertainty of the model. The uncertainty is about an OM and is increased to 2 OMs when all the parameters are coincidentally at their low/high ends:

Line 237-239: “We find that different ensemble members span about an OM in predicting the total surface ocean plastic mass. This variability range is larger than that of the individual parameters as discussed above (about 0.5 OM).”

Line 242-243: “The total surface ocean plastic mass predicted by these two models varies about 2 OMs.”

(30) Page 5 lines 32-37: *these lines are not clear and define the “uncatchable plastic” – consider to rewrite and specify the importance of this part.*

Uncatchable plastics are those with diameter less than 0.33 mm that is the standard mesh size of neuston trawls. So, these plastics are “uncatchable” by regular trawls, resulting in the underestimation of surface plastic concentrations in observations. These lines were modified as:

Line 187-196: “The modeled fraction of uncatchable plastics with diameter less than 0.33 mm that is the standard mesh size of neuston trawls, varies between 0.18% and 18% in the surface ocean. Plastics below this size could not be caught by regular trawls and cause underestimated surface plastic concentrations in the observation dataset. Our model thus could appraise such underestimates. The model also suggests that more uncatchable particles are generated with stronger fragmentation/abrasion. In another scenario, we specifically increase the fragmentation/abrasion rate of microplastics by 10 times (but keep that for macroplastics unchanged), as small plastics degrade faster than the large ones due to a higher surface-area-to-volume ratio (Eriksen et al. 2014). The fraction of uncatchable plastic in the surface ocean also increases to 5.1%.”

(31) Page 6 top: *it is nice to see the results for the biofouling; consider to add a table to summarize the same results for the different scenario/process. How the present results compare to the Kvale et al. study?*

The modeled total surface plastic masses of different scenarios are shown in Fig 3. Perturbing the biofouling rate by an OM results in a 0.47 OM change in total plastic mass in the surface ocean in our model. Kvale et al. 2020 showed that the near-surface microplastic concentrations vary more than 2 OMs under different biofouling scenarios. We added the following sentence in the revised manuscript:

Line 208-209: "... smaller than the results of Kvale et al. (2020) due to the removal effects of zooplankton and marine snow that are highly simplified (represented by the plankton biomass) in this model."

(32) Page 6 table 1: where are used and discussed the "marine sources";

We added the discussions on marine sources in the revised manuscript:

Line 223-231: "Plastic waste from shipping and fishing activities is a non-negligible source for ocean plastics (Richardson et al. 2019; Ryan et al. 2019; Richardson et al., 2022). For example, the increasing number of plastic drink bottles found in the ocean indicates that large amounts of plastic waste originate from ships (Ryan et al. 2019). However, attempts to estimate the magnitude of the marine source are extremely scarce and only a few studies assume that the marine source accounts for 20% of the total plastic discharge (Lebreton et al. 2012; Kane et al. 2020). Not considering the marine source results in a 18% reduction in the total plastic mass on the surface ocean compared with the test case scenario. The surface plastic mass increases by 30% when the fraction of the marine source increases from 20% to 40% (Fig. 3a)."

(33) Page 7 top: How did you explain the difference between the beaching (that vary by 4 for the rate) and the sedimentation rate that vary by 2 OMs?

We thank the reviewer pointing it out. The beaching rate is generally constrained by the fraction of plastics that are transported to the beach, which has an upper bound of 100%. The reported beaching rates vary from 10–94% in three days or 100% in 3–4 weeks (Atwood et al. 2019; Samaras et al. 2014; Hinata et al. 2017; Stanev et al. 2019), i.e., 3.3–31% per day. So, we choose a beaching rate that overlaps with this range (1–25%/day as listed in Table 1, i.e., varies for 1.4 OM). The sediment rate is the settling velocity of plastic particles near the seafloor, which depends on the seafloor flows that can easily vary by 2 OMs (Kane et al. 2020). So, the sedimentation rate is assumed to vary by the same OMs. We modified these sentences as follows:

Line 216-223: "We perturbate the beaching rate from 1% to 25% day⁻¹, which represents two extremes of the behavior of plastic particles, i.e., nearly no beaching and fast capture by beaches, **as suggested by previous empirical studies (Atwood et al. 2019; Samaras et al. 2014; Hinata et al. 2017; Stanev et al. 2019)**. The model suggests that the beached plastics account for 8.1–63% of total discharge for these scenarios (Fig. 3a). A similar level of uncertainty is found for the plastic sedimentation rate, **which could vary as high as 2 OMs as suggested by Kane et al. (2020)**."

(34) Page 7 line 16: the term "potentially" suggests that it may not be the case? Comment on that please.

We believe that the model is able to constrain the magnitude of emissions. The term

“potentially” was deleted.

(35) Page 8 figure 4: consider to improve figure 4; I do not understand what is the y-axis unit? It is not evident to distinguish the 150 orange lines, and the thicker black is corresponding to the 0.72 value, correct? What is the reference to assume that the distribution should be a log-normal one?

The y-axis is for probability distribution function. We changed the diagram to a vector diagram that has much higher quality. The thicker black is for the ensemble mean, i.e., the optimal emission. We added a legend in the figure to make it clear. As shown in Fig S2, the distribution of the modeled plastic mass is a log-normal one so we assume that the distribution of emissions is also a log-normal one. We included a citation of Fig. S2 in the revised manuscript:

Line 270-272: “The curves are the probability distribution function (pdf) of the optimized global plastic emissions assuming a log-normal distribution based on the simulated surface plastic mass (Fig. S2).”

(36) Page 8 line 22: there is no circles on figure S4a, and I do not see any differences with the 2b and 2e parts of figure 2; please clarify and add some comments to help the reader.

We added circles (i.e., observations) on Fig S4a. The best estimate of the global plastic emissions is close to the middle emission scenario. We added the following sentence in the caption of Fig. S4 to help the reader:

“The best-estimate emission is close to the middle emission scenario and the simulated concentrations in panel (b) are just slightly higher than those in Fig. 2e.”

(37) Page 9 line 23: why 54 plastic tracers are considered, and more importantly, what are the impacts of such large number of classes. This point is not discussed at all.

We added a more detailed description of why 54 plastic tracers are considered. These tracers cover the common polymer types, sizes and biofouling states in the ocean:

Line 319-329: “The model includes 5 categories of different chemical compositions and the density of each category is pre-determined: PE (polyethylene, 950 kg m⁻³), PP (polypropylene, 900 kg m⁻³), PVC (polyvinyl chloride, 1410 kg m⁻³), PS (polystyrene, 1050 kg m⁻³), and ABS (acrylonitrile butadiene styrene, 1050 kg m⁻³). Each category has 6 size bins with bounds equally distributed on a log scale: four for microplastic: <0.0781 mm, 0.0781–0.3125 mm, 0.3125–1.25 mm, and 1.25–5 mm, and two for macroplastic: 5–50 mm and >50 mm. The plastic’s density is increased when biofouled and we consider three states for PE and PP: floating, neutral, and sinking (no floating and neutral states for PVC, PS, and ABS as they are originally heavier than or as heavy as seawater). **There are a total of 54 plastic tracers in the model (PE and PP each has 18 tracers while the other three each has 6 tracers), representing the common chemical compositions, size ranges and biofouling states of plastics in the marine environment.**”

(38) Page 9 “general description”: what’s about the settling velocity of the different polymers considered?

Different polymers have different densities that are related to their settling velocity (eq

5). We modified the sentences to make it clearer:

Line 339-341: “Sinking or rising particles make an approximate one-dimensional motion relative to the water column and the velocity depends on their densities and diameters.”

Line 395-396: “The plastic densities that depend on their polymer types, together with their diameters and shapes, and seawater state, determine their sinking/rising velocities.”

(39) Page 10: please define acronyms such as HDI or GDP

Revised as suggested.

(40) Page 10 line 24: does the data proposed by Hoornweg and Bhata-Tata are available?

Jambeck et al. 2015 provided a link to the data proposed by Hoornweg and Bhata-Tata: <https://openknowledge.worldbank.org/handle/10986/17388>. We also included the link to this dataset in the revised manuscript:

Line656-658: “Hoornweg, D., & Bhada-Tata, P. (2012). What a waste: a global review of solid waste management. Urban development series; knowledge papers no. 15. World Bank, Washington, DC, USA. <https://openknowledge.worldbank.org/handle/10986/17388>.”

(41) Page 10 line 38: does the numbers of consumption of different plastic types given for 2013 apply for the whole period?

Yes. There is no available consumption data for the whole period. We clarified this by modifying this sentence as:

Line 384-386: “The discharge of each type of plastic is allocated by its proportion to global consumption in 2013 due to the lack of consumption data for the whole period (China Plastic, 2014) (Table S3).”

(42) Page 11 Eq. 6: it seems that the Cd dependency to Re is based on a specific context; please give the reference or explain how these numbers and relationships have been obtained.

We added a reference that summarizes some Cd-Re relationships (Table 2.1 from Khalaf 2009, listed below) and we chose the functions from Chemical engineers' handbook, R. H. Perry and C. H. Chilton (eds.) because it covers a wide range of Re.

Khalaf, HK., The theoretical investigation of drag coefficient and settling velocity correlations. (2009) In: *the College of Engineering*). Nahrain University.

(43) Page 11 line 22: give more details on the “few techniques”

We added how the \mathbf{w} is solved in the supplementary information:

Line 90-125 in the SI: “The sinking/rising rate of plastic depends on its density. Plastics in our model are treated as spheres. At steady state, the forces acting on plastic particles are balanced:

$$\mathbf{F}_D + \mathbf{F}_g + \mathbf{F}_b = 0 \quad (2)$$

where F_D is vertical dragging force, F_g is gravity, and F_b is buoyancy. These forces are calculated as:

$$\mathbf{F}_g = V_p \rho_p \mathbf{g} \quad (3)$$

$$\mathbf{F}_b = -V_s \rho_s \mathbf{g} \quad (4)$$

$$\mathbf{F}_D = -\frac{1}{2} C_D (Re_s) A_p \rho_s \frac{(\mathbf{w} - \mathbf{w}_s)^3}{|\mathbf{w} - \mathbf{w}_s|} \quad (5)$$

where V_p is the volume of the particle, while V_s is the volume of the particle that is submerged in seawater ($V_p = V_s$ in this case, but $V_p > V_s$ for floating particles with zero sinking/rising velocity relative to the seawater, e.g., unbiofouled PP and PE). C_D is the coefficient of dragging, which is a function of the Reynolds number (Re) of a certain motion of a fluid. A_p is the horizontal sectional area of a particle, ρ_s is the density of seawater, ρ_p is the mean density of a particle, \mathbf{w} is the vertical velocity of the particle, \mathbf{w}_s is the velocity of seawater, and \mathbf{g} is the gravity acceleration.

Based on Equation (2)–(5), we get $(\mathbf{w} - \mathbf{w}_s)^2$:

$$(\mathbf{w} - \mathbf{w}_s)^2 = \frac{4|\mathbf{g}|d(\rho_p - \rho_s)}{3C_D(Re_s)\rho_s} \quad (6)$$

where d is Stokes diameter of a particle, and C_D is calculated as (Khalaf 2009):

$$C_D(Re) = \begin{cases} 24Re^{-1} & Re \leq 0.3 \\ 18.5Re^{-0.6} & 0.3 < Re \leq 1000 \\ 0.44 & 1000 < Re \leq 20000 \end{cases} \quad (7)$$

Re_s is the Re of seawater and is calculated as:

$$Re_s = \frac{d\rho_s |\mathbf{w} - \mathbf{w}_s|}{\mu_s} \quad (8)$$

Based on Equation (6) (7), we get $\mathbf{w} - \mathbf{w}_s$:

$$\mathbf{w} - \mathbf{w}_s = \begin{cases} \frac{d^2 g(\rho_p - \rho_s)}{18\mu} & Re \leq 0.3 \\ \frac{0.153d^{1.143}[g(\rho_p - \rho_s)]^{0.714}}{\rho_s^{0.286}\mu^{0.429}} & 0.3 < Re \leq 1000 \\ 1.74\left(\frac{dg(\rho_p - \rho_s)}{\rho_s}\right)^{\frac{1}{2}} & 1000 < Re < 200000 \end{cases} \quad (9)$$

Equation (6) is a piecewise function. The three conditions of different Re are all possible in reality because of the variety of particles, but it is hard to know which range of (6) should be used in a certain situation, since Re should be calculated directly by sinking velocity \mathbf{w} , which is unknown here.

Notice that (7) can be represented by:

$$C_D = \frac{a}{Re^b} \begin{cases} a = 24, b = 1 & Re \leq 0.3 \\ a = 18.5, b = 0.6 & 0.3 < Re \leq 1000 \\ a = 0.4, b = 0 & 1000 < Re < 200000 \end{cases} \quad (10)$$

Combine (6) (8) (10) and eliminate \mathbf{w} and C_D :

$$Re = \left(\frac{4}{3a}\right)^{\frac{1}{2-b}} \left(d^3 \frac{\rho_s g(\rho_p - \rho_s)}{\mu^2}\right)^{\frac{1}{2-b}} \quad (11)$$

Define K :

$$K = d^3 \frac{\rho_s g(\rho_p - \rho_s)}{\mu^2} \quad (12)$$

Then:

$$Re = \begin{cases} 0.0556K & Re \leq 0.3 \\ 0.072K^{\frac{5}{7}} & 0.3 < Re \leq 1000 \\ 1.74K^{\frac{1}{2}} & 1000 < Re < 200000 \end{cases} \quad (13)$$

$$\mathbf{w} - \mathbf{w}_s = \begin{cases} \frac{d^2 g(\rho_p - \rho_s)}{18\mu} & K \leq 5.4 \\ \frac{0.153d^{1.143} [g(\rho_p - \rho_s)]^{0.714}}{\rho_s^{0.286} \mu^{0.429}} & 5.4 < K \leq 24 \\ 1.74 \left(\frac{dg(\rho_p - \rho_s)}{\rho_s} \right)^{\frac{1}{2}} & K > 24 \end{cases} \quad (14)''$$

(44) Page 11 line 24: “in the seawater column” the term u as a sinking or rising velocity is not usual, I will prefer to see a w . How the different types of polymer is considered in this computation should be more specified and detailed.

The “ \mathbf{u} ” was modified as “ \mathbf{w} ” as suggested in the revised manuscript. The density of a plastic particle, which is determined by polymer type, affects the sinking/rising velocity (eq 5). Also, the sinking/rising velocity is influenced by the diameter of a plastic particle. So, plastic particles with different polymer types and diameters have different sinking/rising velocities. The sentences were modified as:

Line 395-396: “The plastic densities that depend on their polymer types, together with their diameters and shapes, and seawater state, determine their sinking/rising velocities.”

(45) Page 12 lines 20-22: what’s about the surface level? Most of the diffusion process involved a laplacian or a bi-laplacian operator with some constant coefficients. Consider to give the magnitude of these coefficients and to clarify the importance of the “random walk” in such a coarse model.

Our model follows the setting up of Dutkiewicz et al. (2009), which gives the Laplacian operator for the tracers in the MITgcm model:

$$\frac{\partial P}{\partial t} = -\nabla \cdot (\mathbf{u}P) + \nabla \cdot (K\nabla P) + \dots$$

where \mathbf{u} is the velocity in physical model ($\mathbf{u} = u, v, w$) and K is the mixing coefficients used in physical model. The mean diffusion coefficient is $10^{-3} \text{ m}^2 \text{ s}^{-1}$. We clarified this by adding the following sentences:

Line 447-448: “..., which follows the Laplacian operator with the mixing coefficients given in Dutkiewicz et al. (2009).”

(46) Page 12 line 35: specify the year for the reference to Lebreton et al.

This sentence was removed in the revised manuscript.

(47) Page 13 eq 14: Do not understand how this relationship is found; in line 13 the mention of the reference Biber et al. is unclear because these latter authors based their study of different types of polymers, and not to real plastics found in polar or tropical regions; please clarify.

There is no available data about the fragmentation rates in polar or tropical regions. Biber et al. (2019) shows that the degree of plastic degradation varies a lot in different environments. The sentences were modified as follows:

Line 464-471: “Consistent with existing observed data, the total fragmentation and abrasion rate R (% yr^{-1}) in seawater is assumed to be 3-30% yr^{-1} , with 10% yr^{-1} of the total R (i.e., 0.3-3% yr^{-1}) is allocated as the abrasion rate (Niaounakis, 2017). The rate in the surface ocean (R_{surf}) is increased proportionally to the downward shortwave solar radiation (q , also a proxy for temperature) in the surface ocean if the plastic particles are not biofouled, reflecting the **dependence of weathering rate on sunlight revealed by controlled experiments** (Brandon et al. 2016; Biber et al. 2019):

$$R_{\text{surf}} = \frac{R}{175} \cdot q + R \quad (14)''$$

(48) Page 13 line 33: explain why there is a so large variations in the percentage of sandy shorelines

We used the dataset for the fraction of sandy shorelines by Luijendijk et al. (2018). A brief explanation was added in the revised manuscript:

Line 493-496: “The global sandy beaches dataset is from Luijendijk et al. (2018). The global estimated percentage of sandy shorelines varies from 10% to 75% due to the lack of reliable global-scale assessment of occurrence or rates of sandy shoreline change.”

(49) Page 13 line 39-40: the reference to the atmospheric chemistry is not really relevant here.

The sentence was deleted in the revised manuscript.

(50) Page 15 table S6: give the sources or reference of parameters such as mortality rate, algae density or shear rate

These parameters (as listed in Table S6) are from the supplementary information of Kooi et al. (2017). We added the reference in Table S6.

(51) Page 15 line 25: metric?

The “matirc” was modified as “metric” in the revised manuscript.

We also checked the full text and corrected the spelling mistakes

(52) Page 15 eq 24 and line 35: what is the term “a” exactly?

“a” is for the posterior estimation, i.e., analysis, derived from the super ensemble and observations.

(53) SI Fig S1: could we detect an abrupt decrease associated with MARPOL in the computation based on Weiss scenario?

Yes. There is still a decrease associated with MARPOL in the Weiss scenario. But it is suppressed by the choice of the range of y-axis.

(54) SI Fig S4: please clarify as compared to figure 2 in the main text.

The following sentence was added to the caption of Fig. S4:

Line 157-158: “The best-estimate emission is close to the middle emission scenario and the simulated concentrations in panel (b) are just slightly higher than those in Fig. 2e.”

(55) I am not sure if the table S8 is referenced in the main text; please verify. Also, give the status of fibers, if they are considered or not. Does the 194 number for Cozar et al. should be 1943?

Table S8 was referenced in the revised manuscript:

Line 91-93: “We limit the observations to the data obtained by visual identification due to its large sample size and spatial coverage (Law et al. 2010; 2014; Eriksen et al. 2014; Cózar et al. 2014) (Table S8).”

The 5th column of Table S8 “Without fiber (%)” shows whether the fibers are considered or not.

We checked the original reference and the reviewer is correct. The number of data points for Cozar et al. (2014) was thus updated to 1943.

Reviewer #3 (Remarks to the Author):

Review: Plastic waste discharge to the global ocean constrained by seawater observations
This paper attempts to constrain the highly uncertain estimates of plastic emissions into the ocean using observations of floating plastic debris combined with an ocean transport model. The magnitude of plastic emissions into the ocean is an important research question that the community is trying to solve for years. Given the uncertainty in parameterization of the plastic emissions and the lack of direct observations, the author's approach to use observations of floating plastic to constrain the input is very promising. Reducing the uncertainty in the input estimates to 0.25–2.14 mio Mt/yr would be a significant step forward. I have read this paper with great interest.

We thank the reviewer for the recognition of our work and the comments and suggestions for improving this manuscript.

I have two main concerns with this work:

(1) The model described in this work is constrained only using surface ocean plastic observations. In order fit the surface observations, the model will transfer the entire excess plastic towards the remaining reservoirs. There needs to be a comparison with existing observations, or at least a discussion how realistic the amount of plastic in these additional reservoirs is. As an example, models of plastic beaching have shown that the modeled amount of beached plastic is orders of magnitude higher than what is observed (see for instance Onink et al. 2021 (<https://iopscience.iop.org/article/10.1088/1748-9326/abecbd/meta>), Ryan et al. 2020 (http://www.scielo.org.za/scielo.php?pid=S0038-23532020000300003&script=sci_arttext&tlng=es) and Collins et al. 2019 (https://www.sciencedirect.com/science/article/pii/S0025326X18308798?casa_token=Otjls0MuVlsAAAAA:dMTajoiYmt2D6RFBYwpD4rhgu3Mvcog5d7TNsd74OJHF1XUxartVBgcSfXdITNbcP_4IOISdRg)) when using the Lebreton or Jambeck input estimate. This is even the case when considering only 1 year of plastic emissions and using a lower beaching rate than in the Lebreton 2019 box model. I therefore think that the present study likely also overestimates the amount of beached plastic, and potentially also the amount of plastic in other reservoirs.

We thank the reviewer for bringing it up. We totally agree with the reviewer regarding the importance of additional reservoirs. We added a comparison between the simulated vertical profile and the observations (Egger et al. 2020; Pabortsava and Lampitt 2020) in the supplementary information (Figure S5). Generally, our model captures the vertical trend of plastic concentrations both in the North Pacific Ocean and the Atlantic Ocean: high concentrations in the surface water and decreased concentrations with the depth. So, we believe that our model can predict the fraction of plastics in each compartment (e.g., surface, water volume, and beaches). In the Pacific Ocean (Fig. S5a), we did not compare the profiles in the deep ocean because the concentrations under the depth of 500 m are below the detection limit (Egger et al. 2020). In the Atlantic Ocean (Fig. S5b), the observed concentrations are elevated in the subsurface (e.g., 100 – 250 m) in some sites while the simulated concentrations decrease slowly in all sites. Indeed, the vertical distribution of microplastics in the water column is an emerging research field that merits further evaluation and we are considering a more detailed parameterization of this process in our recent studies.

The following discussions were added in the main text:

Line 289-292: “Finally, the observations of plastic abundance in other compartments, e.g., the water column, are rather limited to evaluate the model, but our model captures the observed vertical trend, indicating a reasonable representation of the fraction of plastic mass in the surface ocean (Fig S5).”

Fig. S5. Comparison between observed and modeled vertical profile of plastic mass for the a) Pacific and b) Atlantic Ocean. The dash lines indicate individual observed profiles with the filled-colored circles representing sampling depths. The light blue solid lines are the mean plastic mass concentrations over the sampling sites by the 52 member models under the middle emission scenario (other emission scenarios simulate higher or lower concentrations but with similar vertical trends), while the dark blue solid lines are the ensemble means. The observations are from Egger et al. (2020) and Pabortsave and Lampitt (2020) for the Pacific and Atlantic Ocean, respectively.

As for the beach reservoir, we built a model ensemble that covers a wide range of beaching rate/fraction based on existing observations in the literature, which has a large variability across studies. The model beaching rate was assumed to be 2.5-18% day⁻¹ and the modeled fraction of beached plastics varies between 1.9% and 74%. We consider each member model in the ensemble as equal, i.e., no best or standard model is pre-defined, and we use the ensemble to bracket the possible uncertainties. As long as the uncertainty range of our ensemble encompasses that of the observations, we consider our model as not biased.

(2) I think that the range of at least two crucial model parameters, the fragmentation rate and the beaching rate, has been chosen too narrow and as a result the model may underestimate the full uncertainty of the input emissions. Please find more details in the text below.

We thank the reviewer for this valuable suggestion. We added another two extreme model members into the ensemble (please refer to our response below). These two members have all the parameters sampled coincidentally the lowest/highest within their ranges to make the surface plastic mass the lowest/highest. Among the total 52 members, the fraction of beached plastics varies between 1.9% and 74%, which is a relatively wide range. The fraction of uncatchable plastics (diameter < 0.33 mm that is the standard mesh size of manta trawls) in the surface ocean, which is determined by the fragmentation rates, also varies substantially between 0.18% and 32%. We consider this range already large enough to encompass the real

condition. In addition, increasing the variability range of individual parameters would have similar effect as we include the two extreme members. So, we kept the ranges of individual parameter unchanged but added two extreme members.

(3) *Additionally, the model does not include interaction with zooplankton or aggregates, which may increase the transport out of the surface ocean. There should be a discussion how these missing processes may affect the uncertainty envelope of the input emissions.*

We appreciate the reviewer for this helpful suggestion and this process will be considered in our future work. The following discussions were added in the revised manuscript:

Line 206-209: “Perturbing the biofouling rate by an OM result in a 0.47 OM change in total plastic mass in the surface ocean (Fig. 3a), smaller than the results of Kvale et al. (2020) due to the removal effects of zooplankton and marine snow that are highly simplified (represented by the plankton biomass) in this model.”

I hope this helps! Congratulations on the paper.
Charlotte Laufkötter

Line by line comments:

(4) *Abstract line 30: Please add that the 0.72 mio Mt/yr estimate is for present day (I think?)*

The sentence was modified as follows:

Line 27-30: “Here, we adopt a top-down approach that uses observed seawater plastic data and an ensemble of ocean transport models to reduce the uncertainty of the global plastic discharge to about 1.5 OM: 0.70 (0.13-3.8 as a 95% confidence interval) million metric tons yr⁻¹ **at the present-day.**”

(5) *Abstract Line 31ff: “We find the uncertainty of the estimated global ocean plastic emissions is mainly caused by that of model parameters, indicating that a more accurate measurement of the biogeochemical rates of plastics in the ocean is required.” I think the authors mean “biogeochemical rates of plastic losses from the surface ocean”?*

The biogeochemical rates of plastics mean the rates of transport and transformation of plastic particles in the ocean. The sentence was modified as:

Line 30-33: “We find the uncertainty of the estimated global ocean plastic emissions is mainly caused by that of model parameters, indicating that a more accurate measurement of **the rates of plastic transport and transformation** in the ocean is required.”

(6) *Fig.1: “Lebreton and Andrady (2019) include both riverine and coastal zone sources” - I may be mistaken, but I think the paper follows the Jambeck approach which would be only coastal zone sources. Please double check.*

We double checked and Lebreton and Andrady (2019) considered the riverine sources. In their Methods Section they wrote:

“We categorised grid cells from small watersheds (<10 km²) to continental rivers (>1,000,000 km²) by orders of magnitude of surface area (7 categories in total). The main motivation behind this analysis was to identify the fraction of global plastic waste generated

in coastal areas against **the fraction generated inland that may reach the marine environment via rivers.**”

(7) L66: “covering all the major ocean basins” - True, however the majority of observations is from the North Atlantic and Pacific oceans. There seems to be only one cruise in the entire South Pacific and Indian Ocean.

The sentence was modified as:

Line 67-69: “There are more than 3000 samples of surface net tow data, **mostly located in the centers of gyres in the northern hemisphere** that are found accumulating plastic garbage patches (Cózar et al. 2014).”

(8) L83: Which “important processes” does the model include? Would be good to mention here

The sentence was modified as:

Line 85-87: “which explicitly includes a comprehensive representation of the important processes for plastic particles in the ocean (e.g., sinking and rising, drifting, fragmentation/abrasion, beaching, and biofouling and defouling).”

(9) L87: grammar

The sentence was modified as:

Line 91-93: “We limit the observations to the data obtained by visual identification due to its large sample size and spatial coverage (Law et al. 2010; 2014; Eriksen et al. 2014; Cózar et al. 2014) (Table S8)”

(10) L 91/92 grammar

The sentence was modified as:

Line 96-99: “Mass concentrations are used to compare with the observed surface concentrations as plastic discharge inventories are also in a mass unit, while number concentrations are transformed to mass concentrations as they are prone to larger uncertainties due to the fragmentation processes (Cózar et al. 2014) (see SI Text).”

(11) L93: I am assuming that a substantial uncertainty is added when introducing from numbers to mass. See for instance Roebroek et al. 2022

(<https://www.sciencedirect.com/science/article/pii/S0269749122011629>) where this problem is discussed for riverine plastic observations. Maybe this should be discussed? I don't know what typical values for the number concentrations are in your conversions, the uncertainty is reduced of course with large sample sizes.

In our study, the mean item-mass conversion factor of all observed concentrations is 31 g per item, with the range of 1.3×10^{-5} - 2.4×10^3 g per item. We added a discussion in the Inclusion of seawater observations Section in the supplementary information:

Line 52-55: “The conversion bears uncertainty but the uncertainty is reduced due to the large sampling number (Roebroek et al. 2022). The item-mass conversion factors of all observed concentrations range from 1.3×10^{-5} to 2.4×10^3 g per item, with a mean value of 31 g per item.”

(12) L 112: *The median Jambeck estimate is 8 mio tons, yet the high scenario is only 7.1 mio tons. Was only the lower end of the Jambeck estimates used? If so, why?*

The riverine and coastal input data used in our model is from Chassignet et al. (2021), and they obtained the data from Lebreton and Andrady (2019). Lebreton and Andrady (2019) followed the framework of Jambeck et al. (2015) but had a lower estimation between 3.1 and 8.2 million tonnes. We clarified this in the revised manuscript:

Line 367-369: “The coastal plastic emission inventory follows the framework of Jambeck et al. (2015) that use MPW as proxy data to estimate plastic discharge but **has a lower estimation** (Chassignet et al. 2021).”

(13) *Fig. 2: The match between model and observations does not seem particularly strong, with often a mismatch of 2 orders of magnitude (same for Fig. S4). The R2 value is not very high, there are no other metrics given. I am assuming that I am looking at a point cloud consisting of 5000 measurements with most of the dots overlapping. It's hard to judge how good the model/data fit is. Please add at least the bias.*

There are a total of 764 circles in the Fig 2 as we grouped the observations within each $2.5^\circ \times 2^\circ$ -degree grid box (otherwise, as the reviewer pointed out, there will be too many points to show). For the middle scenario, 78% of the circles have a mismatch lower than 1 order of magnitude. We also added the root mean square error (RMSE) in the figure.

(14) *Also Fig. 2 and S4: I am not sure why the color doing showing the latitude is of interest here - there does not seem to be a pattern with certain latitudes matching better than others?*

We used the same color for all circles in the revised manuscript.

(15) *I also wonder: The overall mass of floating plastic debris is estimated to be 14-236 thousand tons. How was this uncertainty included in the model design / optimization?*

These estimations are purely based on the observed surface ocean plastic abundance and the uncertainty arises from the inherent variability of these observed data (Cózar et al. 2014; Van Sebille et al. 2015, Eriksen et al. 2014). Our method, using an inverse approach, also takes into account all these data and their uncertainties (i.e., the second term in the right-hand side of eq. 24). Since the posterior emission estimate represents the solution of the maximal likelihood (minimum variance), the uncertainty of posterior emission ($\sigma_E^{a,2}$) is smaller than that of either the prior emission (σ_E^2) or observations ($\sigma_{o,p}^2$), given by

$$\frac{1}{\sigma_E^{a,2}} = \frac{1}{\sigma_E^2} + \sum_{p=1}^P \frac{1}{\sigma_{o,p}^2}$$

We also modified the revised manuscript accordingly:

Line 576-579: “The observation error variance is the variance of observed quantities within a model grid for each observation, **which covers the uncertainty of previous estimations of the overall floating plastic mass based on the observed data** (Cózar et al. 2014; Van Sebille et al. 2015, Eriksen et al. 2014).”

(16) L 127-133: *Please add an observational estimate for the amount of floating plastic in the respective regions.*

We modified the sentences as follows:

Line 133-143: “Taking the Middle scenario as an example, the “test case” model (Table1; see Methods for more details) calculates that the concentrations of plastic accumulating in these regions range between 0.2 and 4.4 kg km⁻² while the observed range is <0.01-16 kg km⁻². Compared with the northern hemisphere, accumulation zones in the South Pacific Ocean and South Atlantic Ocean present lower modeled concentrations (0.2-1.4 kg km⁻² vs. <0.01-2.3 kg km⁻² in observation) due to the lower terrestrial discharge. The model allows us to identify additional zones of plastic accumulation. Heavy discharge from Asia results in 0.2-6.1 kg km⁻² in the western subtropical Pacific Ocean (especially the coastal regions) and the highest observed concentration is 14 kg km⁻². Simulated concentrations near the discharge points along the coastlines of South Asia and Europe are also relatively high (up to 5.1 kg km⁻² vs. 3.1 kg km⁻² in observation).”

(17) L 156: *Please clarify that this sentence does not describe the model but the current state of knowledge.*

Revised as suggested. The sentence was modified as:

Line 182-183: “**In the previous studies**, the fragmentation/abrasion rates are a function of environmental factors including light, temperature, oxygen, and plankton biomass (Andrady 2011).”

(18) L 156ff: *I appreciate these sensitivity tests! I do believe though that the fragmentation rates have been chosen too high (or maybe that a scenario with lower fragmentation rates should be added). See comments below.*

We thank the reviewer for the appreciation! We added two extreme scenarios with all the parameters sampled coincidentally the lowest/highest within their ranges into the ensemble, which increases the variability range across the ensemble members (please refer to our response below). We believe that this modification makes us better encompass the real conditions. As for the fragmentation rate, the existing range of fragmentation rates has already largely changed the model results. Further enlarging the variability of this rate should have a similar effect as the two extreme scenarios we added. Please refer to our response to the 2nd major comment.

(19) L 186: *The Lebreton 2019 and the Chenillat papers are cited as estimates for the amount of plastic at beaches, almost like a model evaluation. However, I believe the Lebreton paper does not actually show that: The Lebreton study assumes a high plastic input and optimizes a simple box model such that the model matches the amount of floating surface plastic, similar to what has been done in this study. In order to match the surface plastic concentration, the model has no other option than to transport the remaining plastic to beaches or sediments, but without checking if that is a realistic outcome. The Chenillat paper likewise does not evaluate their modeled beaching results. In addition, the beaching parameterization they have used is unrealistically high - see discussion in Onink et al. 2021. Based on the results in*

Onink et al. 2021 and also Ryan et al. 2020 and Collins et al. 2019 I believe that these models severely overestimate beaching of plastic. See also comments below.

We thank the reviewer for this insightful consideration. We removed the comparison with Lebreton et al. (2019) and Chenillat et al. (2021) in the revised manuscript. We allocated a wide range of values for each parameter, including the beaching rate to contain the real condition in the study. The beaching rate varies from 1% to 25% day⁻¹ in the sensitivity experiments and the resulted beached plastic account for 8.1-63% of total discharge. In the model ensemble, the 95% confidence interval for the beaching rate is 2.5-18% day⁻¹. The resulted beached plastic account for 1.9% and 74% of total discharge, which could encompass the ranges reported by later studies the reviewer mentioned (e.g., Onink et al. 2021; Ryan et al. 2020; Collins et al. 2019). As different ensemble members are considered equal in the ensemble, i.e., no pre-defined standard model, and the range of the beaching rate/fraction is wide enough to contain the real condition, we thus kept the range of the beaching rate unchanged in the revised manuscript. In addition, two more extreme member models were added to the ensemble (please refer to our response below), which further expands our parameter ranges and increases the chance of our ensemble to contain the real conditions.

(20) L 201: Given the large uncertainty associated with almost all model parameters and also the fact that some processes (such as interaction with zooplankton or marine aggregates) are missing in the model, I am not convinced that it is so unlikely that all parameters are at the higher or lower end. It would at least be important to know how large the collective variability is. Please add scenarios with all parameters assuming highest/lowest values.

We agree with the reviewer and thank for this great suggestion. We added member 51 and member 52 in our ensemble (Fig. 3b). The parameters of #51 (e.g., the lowest marine source and the highest beaching rate) lead to the lowest surface plastic mass while the parameters of #52 (at the opposite end of #51 for all parameters in their ranges) lead to the highest surface plastic mass. The variation in surface plastic mass for all the members is increased from 0.92 OM_s for #1-50 to 1.97 OM_s for #1-52. The following sentences were added in the revised manuscript:

Line 240-243: “We also include another two models (#51 and #52) that sample the lowest/highest parameters to represent the largest possible collective variability and the surface mass range. The total surface ocean plastic mass predicted by these two models varies about 2 OM_s.”

Line 555-558: “We also include another two models (#51 and #52) that sample the lowest/highest parameters. The parameters of #51 (e.g., the lowest marine source and the highest beaching rate) lead to the lowest surface plastic mass while the parameters of #52 (at the opposite end of #51 for all parameters in their ranges) lead to the highest surface plastic mass.”

(21) L 408: Both the Song paper and the Efimova paper describe fragmentation of plastic on beaches, which is not directly transferable to open ocean conditions. The open ocean fragmentation is likely lower.

The Song and the Efimova papers were replaced by Welden and Cowie (2017) and

Gerritse et al. (2020). The sentences were modified as:

Line 458-464: “The rate of fragmentation in seawater has limited data but varies drastically depending on the type and shape of plastics, as well as the environmental factors (Welden and Cowie, 2017; Gerritse et al. 2020). For example, PE and PP lose 0.39-1.02%, with an average of 0.45% and 0.39%, of their mass per month, respectively (Welden and Cowie, 2017). In another laboratory experiment, seawater PE, PP, and PS lose $\leq 1\%$ of their weight per year while the ratio of loss is higher (3-27%) for other polymers such as polyurethane and polyester (Gerritse et al. 2020).”

(22) L413: *The Lebreton model optimizes the observed surface plastic concentration when using a given plastic emission estimate, similarly to what is done in this work. I don't think this study can be used to constrain the fragmentation rate, what they report is the most likely fragmentation rate assuming a certain plastic emission. If the plastic emission they are using is off or likewise any of the other fluxes they assume such as the beaching, their numbers would change.*

We removed the reference of Lebreton et al., 2019. We developed a model ensemble in this study and the fragmentation and abrasion rate are assumed to vary between 3% and 30% yr^{-1} as ± 1 standard deviation. Note that random sampling may pick up the values outside this range but the probability is relatively low. Similar to the logic for our response to the comment #19, we think our relatively large range of the 52-member ensemble could well contain the real conditions.

The sentences were modified as:

Line 464-471: “Consistent with existing observed data, the total fragmentation and abrasion rate R ($\% \text{ yr}^{-1}$) in seawater is assumed to be 3-30% yr^{-1} , with 10% yr^{-1} of the total R (i.e., 0.3-3% yr^{-1}) is allocated as the abrasion rate (Niaounakis, 2017). The rate in the surface ocean (R_{surf}) is increased proportionally to the downward shortwave solar radiation (q , also a proxy for temperature) in the surface ocean if the plastic particles are not biofouled, reflecting the dependence of weathering rate on sunlight revealed by controlled experiments (Brandon et al. 2016; Biber et al. 2019):

$$R_{\text{surf}} = \frac{R}{175} \cdot q + R \quad (14) ”$$

(23) L 422: *So what values are assumed for R? I am assuming that q is in W/m2 and varies between 0-800 W/m2 roughly, that would lead to R being between 10-55 %/yr. This seems like an unreasonably high fragmentation rate for the open ocean. See also Onink et al. 2022 (just published): <https://pubs.acs.org/doi/10.1021/acs.est.2c03363>*

R is the total fragmentation and abrasion rate in seawater and is assumed to be 3-30% yr^{-1} following a log-normal distribution. The q in our model varies between 0-470 W m^{-2} so the R_{surf} is about 3-110% yr^{-1} . The resulted fragmentation time scale is 0.9-33 years, which is well within the range suggested by Onink et al. (2022).

(24) L 428: “A plastic particle is beached when it arrives at a beach-adjacent cell”. - Do you mean they always beach once they are in a beach-adjacent cell, or that there is a probability that they beach according to the beaching rate described later?

As we show later in the Beaching Section, a portion of the plastic particles in a beach-adjacent cell is beached and the portion depends on the beaching rate and the fraction of sandy beaches in this cell. The sentence was modified as:

Line 474-475: “A plastic particle has a chance to be ‘beached’ (i.e., deposited onto beaches) when it arrives at a beach-adjacent cell. The chance or ‘beaching rate’ depends on ...”

(25) L 431: *The Lebreton paper is not a good citation here: They assume that beached plastic resuspends into the ocean, but they don’t actually show it. Better to cite one of the Hinata papers here.*

Revised as suggested.

(26) L 432: *Please mention that the Atwood paper is about microplastic (I think)*

Revised as suggested. The sentence read as follows in the revised manuscript:

Line 478-479: “Atwood et al. (2019) found that <10–94% of released **microplastics** are beached and the majority of beaching occurs within the first three days, ...”

(27) L 433 *It is unclear to me how you transformed the drifter data into a beaching rate. In what distance from the shore were the drifters floating, and how long did it take until they beached?*

We got the beaching rate based on the fraction of beached drifters over the total number of releases in a given period. For example, ocean drifter studies reveal that the timescales of the beaching and resuspension processes range from three to four weeks under different conditions (Samaras et al. 2014; Hinata et al. 2017; Stanev et al. 2019). These timescales can be transferred to beaching rates between 100% per 21 day and 100% per 28 day. We added the following sentence to clarify this point:

Line 478-480: “Atwood et al. (2019) found that <10–94% of released microplastics are beached and the majority of beaching occurs within the first three days, which is about 3.3-31% day⁻¹.”

Line 480-484: “Ocean drifter studies reveal that the timescales of the beaching and resuspension processes range from three to four weeks under different conditions (Samaras et al. 2014; Hinata et al. 2017; Stanev et al. 2019). These timescales can be transferred to beaching rates between 100% per 21 days and 100% per 28 days, i.e., 3.6-4.8% day⁻¹.”

(28) L 436: *If you assume a net beaching rate (i.e. beaching - resuspension) as you describe, you simulate beaches as a final sink for plastic. You need to double-check if the simulated overall amount of plastic on beaches is realistic. How many kg of plastic / m2 does your model suggest? See for instance Onink et al. 2021 for a comparison of simulated and observed beached plastic.*

We thank the reviewer for bringing up this point. We totally agree that a multiple environmental reservoir (or metric) model is required to capture the full picture of environmental plastics, which is also our future model development directions. In our current model, we assume a part of plastics in beach-adjacent cells are permanently removed from the oceans. This model can only predict the mass of beached plastics in the unit of kg km⁻²,

so it is not able to be directly compared with the observed beached plastic concentrations (often in the unit of kg km^{-1}). However, as we mentioned in our response to comment #19, our ensemble well brackets the observed net fraction of beaching plastics. We acknowledged this drawback and discussed the future research possibility in the revised manuscript:

Line 292-293: “Evaluation of plastics stored in multiple reservoirs is important and merits further investigation.”

(29) L 440: *Why is 10%/day a conservative beaching rate? It seems to be one possible beaching rate among many. Also, I understand that the model resolution is roughly 2 degrees, so a plastic particle entering a coastal grid cell may have to travel 200km to reach the coast, or 100km from the center of the cell. Assuming that 10% of all particles manage to reach the coast per day seems like a particularly high beaching rate to me.*

We thank the reviewer to point it out. We removed the word “conservative” in our revised manuscript. In addition, the effective beaching rate in a grid cell is calculated as $2.5\text{-}18\% \times \text{beach fraction}$. The mean fraction of sandy beaches in all coastal grid cells is 5.9%, so the mean effective beaching rate used in our model varied between $2.5\% \times 5.9\% = 0.15\%/day$ and $18\% \times 5.9\% = 1.1\%/day$, which we think is rather reasonable. The sentences were modified as follows:

Line 488-494: “We assume a net beaching rate in a cell as $2.5\text{-}18\% \text{ day}^{-1}$. Their dynamics of beaching and resuspension are highly variably on local scales. Not all sandy beaches have a net accumulation of plastic debris, and not vice versa. We thus consider the fraction of sandy beaches as a proxy for the coastal morphological features. So, the effective beaching rate is corrected by the portion of sandy beaches in a cell, which results in a rate between 0.15 day^{-1} and $1.1\% \text{ day}^{-1}$. This result is also generally consistent with the above-mentioned studies.”

(30) *Supplement Line 66: The estimates of coastal dumping only cover macroplastic (or potentially secondary microplastic), not tire wear particles or lost resin pellets. The Jambeck paper considers mismanaged municipal solid plastic waste, which does not include microplastic.*

We removed the coastal source in the revised manuscript.

(31) *Fig S1: Please explain what you mean with “the abrupt decrease in the 1980s is associated with the MARPOL convention”*

Update: I found the explanation later in the text, maybe just refer to it here

Revised as suggested. Now the sentence reads:

Line 134-135: “The abrupt decrease in the 1980s is associated with the MARPOL Convention that bans the dumping of waste from ships.”

Reviewer #1 (Remarks to the Author):

This paper proposes a method to determine more accurately the amount of plastic discharge to the ocean, based on ocean transports models run for a large ensemble of model parameters controlling the different processes affecting plastic particles (vertical velocity, drifting, fragmentation, beaching, biofouling).

One discharge scenario among the 3 tested provides much better agreement with the available measurements of plastic concentrations at the ocean surface, reducing the current uncertainty about the annual plastic discharge by more than one order of magnitude.

The aim of the paper is obviously interesting.

The method is clearly new in this field, original and appropriate. It is clearly described, and some limitations are discussed.

Its implementation represents an enormous amount of work.

Following the revision of the manuscript, the authors have done a great job in addressing the comments I made on the first version of the manuscript.

The comparison with observations in the water column (Fig. S5) is definitely a nice improvement even given the dispersion of the observed values.

However, when including the Stokes drift, as described in the Supplementary Materials page 3, I understand the annual mean values have been used, and I am just surprised the climatological (1990-2015) seasonal cycle has not been used instead. In fact, I am surprised of the little effect this contribution has on the final surface concentrations.

I think the paper could now be accepted for publication.

There are some minor points I have found in the Supplementary Materials that should be addressed in the final version but do not need another revision:

Fig. S1. Maybe consider plotting in logarithmic scale to make Weiss trend visible?

Fig. S5. In the legends of the 2 panels, I believe 'Assemble mean' should be 'Ensemble mean'?

Fig. S6. There is an inconsistency between the caption and the labels of the panels. I am quite surprised there is no more difference between the two experiments. The exact characteristics of the simulation shown here should be given in the caption.

page 5 line 164 : I understand Brach et al. reference has been added in response to the other reviewer's comment. Let me just point out here that the robustness of their measurements as a general result about anticyclones/cyclones discrepancy might be discussed, as pointed out by Vic, C., S. Hascoët, J. Gula, T. Huck, C. Maes, 2022: Oceanic mesoscale cyclones cluster surface Lagrangian material. *Geophysical Research Letter*, 49, (4) e2021GL097488, doi: 10.1029/2021GL097488 .

Reviewer #2 (Remarks to the Author):

Dear colleagues,

This is my second review of your work and I think that your manuscript gains in clarity, and that the main points on the previous version have been taken into consideration; I am pleased to see that you insert the role of small scale into the model uncertainties part, but my idea was to discuss in deep the role of mesoscale eddies, or at least, reveal the absence of such features in your model; it is not evident that the mention only to the Brach et al reference is useful, a work that has been challenged recently by Vic et al (GRL 2022). Another minor point concerns the last sentence of the abstract (mitigation strategies) that is not really discussed in the rest of your

manuscript; please consider to remove or rephrase it.
I am pleased to recommend the publication of your work.

Reviewer #3 (Remarks to the Author):

2nd Review: "Plastic waste discharge to the global ocean constrained by seawater observations"

The authors have answered most of my comments and the revised manuscript is substantially improved. In particular the comparison with deep ocean plastic measurements strengthens their results.

Unfortunately, 2 of my main concerns still remain:

Regarding the fraction of beached plastic: The authors write that the modeled fraction of beached plastic varies between 1.9% and 74% in their results. I don't know to which input scenario the 74% belong, but even when assuming it's the lower end (130.000 tons/yr), that would add ~97.500 tons of plastic to the world's beaches every year (or substantially more if it belongs to the higher input scenario). Is that realistic?

They also write in response to comment 28: "However, as we mentioned in our response to comment #19,

our ensemble well brackets the observed net fraction of beaching plastics." - Please note that the cited studies in #19 are model studies, not observations. And in the cited studies a comparison with observations was attempted, and these models severely overestimate the amount of beached plastic (by orders of magnitude). So, if the ensemble encompasses the net fractions of beached plastic found in other modeling studies, then the amount of beached plastic is very likely drastically overestimated, and hence the authors may overestimate the possible plastic input into the ocean.

At the very least, please add the modeled relative amount of beached plastic to the paper (ideally, the modeled amount of plastic in all model compartments), so that in the future when more observations are available the model results can be put into context.

Regarding the chosen range for the fragmentation rate: The lowest/highest fragmentation rate according to the studies that the authors cite is <1% - 27% mass loss per year, depending on polymer. For the most common polymers (PE and PP), the mass loss is at the lower end (<1%). Based on these numbers, I think the authors should vary the fragmentation rate from <1 to at most 27%, and the 27% is likely substantially too high (as it is only found for some less common polymers).

However, the authors vary the fragmentation rate between 3 and 30%, and at the surface between 3-110%. To me, this sounds like they are strongly overestimating the fragmentation rate, and hence potentially overestimate the annual plastic input.

Note: *italic fonts are the reviewers' comments; blue fonts are our responses with purple for quoted text from the revised manuscript.*

Reviewer #1 (Remarks to the Author):

(1) This paper proposes a method to determine more accurately the amount of plastic discharge to the ocean, based on ocean transports models run for a large ensemble of model parameters controlling the different processes affecting plastic particles (vertical velocity, drifting, fragmentation, beaching, biofouling). One discharge scenario among the 3 tested provides much better agreement with the available measurements of plastic concentrations at the ocean surface, reducing the current uncertainty about the annual plastic discharge by more than one order of magnitude. The aim of the paper is obviously interesting. The method is clearly new in this field, original and appropriate. It is clearly described, and some limitations are discussed. Its implementation represents an enormous amount of work. Following the revision of the manuscript, the authors have done a great job in addressing the comments I made on the first version of the manuscript. The comparison with observations in the water column (Fig. S5) is definitely a nice improvement even given the dispersion of the observed values.

We thank the reviewer for the recognition of our work and the comments and suggestions for improving this manuscript.

(2) However, when including the Stokes drift, as described in the Supplementary Materials page 3, I understand the annual mean values have been used, and I am just surprised the climatological (1990-2015) seasonal cycle has not been used instead. In fact, I am surprised of the little effect this contribution has on the final surface concentrations.

We indeed used the climatological seasonal cycle of Stokes drift in the model. We are sorry for our miscommunication. We clarified this in the revised supplementary information.

SI Line 88-91: “The data are available between 1990 and 2015 with a time resolution of three hours. We calculate a 26-year average of the Stokes drift velocity for each month. In this way, 12 months of Stokes drift velocity is achieved and cycled in the model for the whole simulation period.”

Stokes drift has a small effect on the surface plastic pattern in our model. More work will be done to evaluate the contribution of Stokes drift in our future studies, e.g., using a modeling framework with higher resolutions and more native parameterization of this process.

(3) I think the paper could now be accepted for publication.

We thank the reviewer again for the recognition of our work.

There are some minor points I have found in the Supplementary Materials that should be addressed in the final version but do not need another revision:

(4) Fig. S1. Maybe consider plotting in logarithmic scale to make Weiss trend visible?

Revised as suggested.

(5) Fig. S5. In the legends of the 2 panels, I believe 'Assemble mean' should be 'Ensemble mean'?

The words “Assemble mean” were modified as “Ensemble mean”.

(6) Fig. S6. There is an inconsistency between the caption and the labels of the panels. I am quite surprised there is no more difference between the two experiments. The exact characteristics of the simulation shown here should be given in the caption.

We made the caption consistent with the labels of the panels and clarified that there are differences in the spatial pattern between the two panels in the caption.

SI Line 188-190: “Supplementary Fig. 6: The modeled spatial pattern of surface plastic abundance. a Without Stokes drift. b With Stokes drift. Compared with panel a, Stokes drift slightly changes the spatial pattern in panel b.”

We also included the following sentences in the SI text:

SI Line 91-95: “The strongest Stokes drift is simulated in the Southern Ocean and the high-latitude ocean in the northern hemisphere where plastic concentrations are relatively low (Supplementary Fig. 6), consistent with previous results (Carrasco et al. 2014). The modeled plastic accumulation in the subtropical gyres is moved westward slightly by Stokes drift, while plastics in the Southern Ocean are moved eastward.”

(7) page 5 line 164: I understand Brach et al. reference has been added in response to the other reviewer's comment. Let me just point out here that the robustness of their measurements as a general result about anticyclones/cyclones discrepancy might be discussed, as pointed out by Vic, C., S. Hascoët, J. Gula, T. Huck, C. Maes, 2022: Oceanic mesoscale cyclones cluster surface Lagrangian material. *Geophysical Research Letter*, 49, (4) e2021GL097488, doi: 10.1029/2021GL097488.

The discussions on the effects of eddies were modified as:

Line 143-145: “The effects of some small-scale processes such as anticyclonic and cyclonic eddies are not included in the model. These processes are quite complicated and are found to contribute inversely in different studies (Brach et al. 2018; Vic et al. 2022).”

Reviewer #2 (Remarks to the Author):

Dear colleagues,

(1) This is my second review of your work and I think that your manuscript gains in clarity, and that the main points on the previous version have been taken into consideration.

We thank the reviewer for the recognition of our work and the comments and suggestions for improving this manuscript.

(2) I am pleased to see that you insert the role of small scale into the model uncertainties part, but my idea was to discuss in deep the role of mesoscale eddies, or at least, reveal the absence of such features in your model; it is not evident that the mention only to the Brach et al reference is useful, a work that has been challenged recently by Vic et al (GRL 2022).

The discussions on the effects of eddies were modified as:

Line 143-145: “The effects of some small-scale processes such as anticyclonic and cyclonic eddies are not included in the model. These processes are quite complicated and are found to contribute inversely in different studies (Brach et al. 2018; Vic et al. 2022).”

(3) Another minor point concerns the last sentence of the abstract (mitigation strategies) that is not really discussed in the rest of your manuscript; please consider to remove or rephrase it.

The sentence was removed.

(4) I am pleased to recommend the publication of your work.

We thank the reviewer again for the recognition of our work.

Reviewer #3 (Remarks to the Author):

2nd Review: “Plastic waste discharge to the global ocean constrained by seawater observations”

(1) The authors have answered most of my comments and the revised manuscript is substantially improved. In particular the comparison with deep ocean plastic measurements strengthens their results.

We thank the reviewer for the recognition of our work and the comments and suggestions for improving this manuscript.

Unfortunately, 2 of my main concerns still remain:

(2) Regarding the fraction of beached plastic: The authors write that the modeled fraction of beached plastic varies between 1.9% and 74% in their results. I don't know to which input scenario the 74% belong, but even when assuming it's the lower end (130.000 tons/yr), that would add ~97.500 tons of plastic to the world's beaches every year (or substantially more if it belongs to the higher input scenario). Is that realistic?

They also write in response to comment 28: “However, as we mentioned in our response to comment #19, our ensemble well brackets the observed net fraction of beaching plastics.” - Please note that the cited studies in #19 are model studies, not observations. And in the cited studies a comparison with observations was attempted, and these models severely overestimate the amount of beached plastic (by orders of magnitude). So, if the ensemble encompasses the net fractions of beached plastic found in other modeling studies, then the amount of beached plastic is very likely drastically overestimated, and hence the authors may overestimate the possible plastic input into the ocean.

The highest modeled fraction of beached plastic is constantly 74% for the three emission scenarios and is given by member#31 in the ensemble. The measured concentrations of beached plastics vary widely (Onink et al., 2021), so we use a wide range of beaching rates to bracket the potentially highly variable beached fraction. The beaching rate follows a normal distribution. The fraction of 74% is an extremely high case, which is highly unlikely to happen. The figure below shows the modeled fraction of the 52 members, with a median value of 44%. Also, 40 out of 52 members, accounting for 77% of the ensemble, simulate beached fraction of $\leq 50\%$. So, although several model members simulate high beached fractions, they play a relatively small role in estimating optimal emissions. We added the discussions on this uncertainty in the revised manuscript.

Line 251-261: “Fourth, the observations of plastic abundances in other compartments, e.g., water column and beaches, are rather limited to evaluate the model. Our model captures the observed vertical trend, indicating a reasonable representation of the fraction of plastic mass in the surface ocean (Supplementary Fig. 5). **The model ensemble yields a wide range of fractions of the beached plastics to the total discharge (Supplementary Fig. 7), which brackets the potentially highly variable beached fraction in the real ocean. The high end of the range, which is less likely to occur in the ensemble, is possible to be higher than the observations (Onink et al. 2021; Ryan 2020; Collins et al. 2019).** We thus suggest that a combination of both bottom-up and top-down approaches, such as developing more accurate emission inventories, obtaining more data **for the abundances of plastics in**

seawater and other compartments, and measuring more accurate model parameters, would be the future research directions.”

(3) At the very least, please add the modeled relative amount of beached plastic to the paper (ideally, the modeled amount of plastic in all model compartments), so that in the future when more observations are available the model results can be put into context.

We added a figure that shows the modeled amount of plastics in water column, sediment, and beach of the 52 members in the revised supplementary information. The parameters of the model members are listed in Table S7. Member#12 simulated the lowest beached fraction (1.9%) while member#31 simulated the highest beached fraction (74%). Note that the highest beached fraction does not mean the highest amount of beach plastics as the ratio of marine sources varies.

Supplementary Fig. 7: Modeled amount of plastics in the water column, sediments, and beaches. The 52 ensemble members are driven by the middle emission scenario. The parameters of the model members are listed in Supplementary Table 7.

(4) Regarding the chosen range for the fragmentation rate: The lowest/highest fragmentation rate according to the studies that the authors cite is <1% - 27% mass loss per year, depending on polymer. For the most common polymers (PE and PP), the mass loss is at the lower end (<1%).

Based on these numbers, I think the authors should vary the fragmentation rate from <1 to at most 27%, and the 27% is likely substantially too high (as it is only found for some less common polymers).

However, the authors vary the fragmentation rate between 3 and 30%, and at the surface between 3-110%. To me, this sounds like they are strongly overestimating the fragmentation rate, and hence potentially overestimate the annual plastic input.

Similar to the beaching rate, the fragmentation rate follows a log-normal distribution. The values near the low and high ends are less likely to be picked by the ensemble. At the surface, only when the shortwave radiation is very strong can the fragmentation rate reaches the high end, and the possibility is quite low. We thus think the influence of these extreme values on our overall ensemble could be less significant.

We agree with the reviewer that the fragmentation rate of PE and PP is low and may be lower than 1% per year. Since some studies show that the fragmentation rate of PE and PP is a little higher than 1% per year, e.g., 0.45% and 0.39% per month (Welden and Cowie, 2017) and the fragmentation rate of other types of plastic is higher, we choose 3% per year as a bulk low end for all plastics. Admittedly, it is possible that our model overestimates the fragmentation rate for the most durable plastics. We hope to apply different fragmentation rates to polymers based on their chemical compositions in the future studies if more measured data are available. The sentences about how we choose the fragmentation rate were modified as:

Line 424-431: “For example, in a laboratory experiment, seawater PE, PP, and PS lost $\leq 1\%$ of their weight per year while the ratio of loss was higher (3-27%) for other polymers such as polyurethane and polyester (Gerritse et al. 2020). In another study, PE and PP were found to have higher fragmentation rates $1\% \text{ yr}^{-1}$. They lost 0.39-1.02%, with averages of 0.45% and 0.39%, respectively, of their masses per month (Welden and Cowie, 2017). Considering the various fragmentation rates of different types of plastics, the total fragmentation and abrasion rate R ($\% \text{ yr}^{-1}$) in seawater for all plastics is assumed to be 3-30% yr^{-1} , following a log-normal distribution. Also, 10% yr^{-1} of the total R (i.e., 0.3-3% yr^{-1}) is allocated as the abrasion rate (Niaounakis, 2017).”